# Skin Lesion Classification and Detection Using Machine Learning Techniques: A Systematic Review

**DOI:** 10.3390/diagnostics13193147

**Published:** 2023-10-07

**Authors:** Taye Girma Debelee

**Affiliations:** 1Ethiopian Artificial Intelligence Institute, Addis Ababa 40782, Ethiopia; tayegirma@gmail.com; 2Department of Electrical and Computer Engineering, Addis Ababa Science and Technology University, Addis Ababa 16417, Ethiopia

**Keywords:** skin, cancer, skin disease, skin cancer, melanoma, machine learning, deep learning, detection, segmentation, classification

## Abstract

Skin lesions are essential for the early detection and management of a number of dermatological disorders. Learning-based methods for skin lesion analysis have drawn much attention lately because of improvements in computer vision and machine learning techniques. A review of the most-recent methods for skin lesion classification, segmentation, and detection is presented in this survey paper. The significance of skin lesion analysis in healthcare and the difficulties of physical inspection are discussed in this survey paper. The review of state-of-the-art papers targeting skin lesion classification is then covered in depth with the goal of correctly identifying the type of skin lesion from dermoscopic, macroscopic, and other lesion image formats. The contribution and limitations of various techniques used in the selected study papers, including deep learning architectures and conventional machine learning methods, are examined. The survey then looks into study papers focused on skin lesion segmentation and detection techniques that aimed to identify the precise borders of skin lesions and classify them accordingly. These techniques make it easier to conduct subsequent analyses and allow for precise measurements and quantitative evaluations. The survey paper discusses well-known segmentation algorithms, including deep-learning-based, graph-based, and region-based ones. The difficulties, datasets, and evaluation metrics particular to skin lesion segmentation are also discussed. Throughout the survey, notable datasets, benchmark challenges, and evaluation metrics relevant to skin lesion analysis are highlighted, providing a comprehensive overview of the field. The paper concludes with a summary of the major trends, challenges, and potential future directions in skin lesion classification, segmentation, and detection, aiming to inspire further advancements in this critical domain of dermatological research.

## 1. Introduction

The evolution of machine learning techniques has impacted many sectors. For instance, breast cancer detection and classification [1,2], diabetes detection and prediction [3,4], and brain tumor detection and classification [5,6] are some of the impacts that machine learning techniques have shown in the health sector in the past few years [7]. The agricultural sector [8,9] and financial sector [10] are also sectors that have benefited from machine learning techniques. In recent years, we have seen significant advancements in dermatology as researchers and clinicians try to understand the complexities of various skin conditions. Skin conditions affect millions of people worldwide and have a substantial impact on both physical health and quality of life. The skin protects our inside organs from microbes, regulates temperature, and serves as a sensation organ [11]. Human skin has three layers: epidermis, dermis, and hypodermis [12]. The epidermis is the outermost layer of skin, which provides a waterproof barrier. It creates our skin tone as it contains special cells called melanocytes, which produce the pigment melanin. The dermis is found under the epidermis and contains tough connective tissues, hair follicles, and sweat glands. The hypodermis is made of fat and connective tissue. Any of the diseases or disorders that harm these layers of skin can be categorized as skin disease.

Apart from the disability and morbidity caused by skin diseases, skin cancer can be fatal if not treated early. Skin cancer occurs when abnormal cells grow uncontrollably in the skin [13]. According to the American Cancer Society [14], 1.9-million new cancer cases are expected to be diagnosed in 2021. The death rate due to skin cancer in America, during 2021, is predicted to be 1670 deaths per day. Kumar et al. [15] claimed that skin cancer is the most-common cancer in developing countries with the most-advanced diagnostics and prognosis. It has recorded 500,000 new cases in U.S., and that made it be classified as the 19th most-common cancer globally.

It is one of the three most-dangerous and -rapidly expanding cancer types, making it a significant public health issue [16]. One out of every three cancer diagnoses is related to skin cancer, according to the World Health Organization, and the Skin Cancer Foundation reports that the prevalence of skin cancer is rising globally [17]. Both benign and malignant skin tumors can develop from DNA damage caused by exposure to UV light, which results in unregulated cell growth, according to Hasan et al. [18]. Despite their growth, benign tumors do not spread. They include pyrogenic granulomas, cysts, cherry angiomas, seborrheic keratosis, dermatofibromas, skin tags, and dermatofibroma. Malignant tumors, on the other hand, can invade other tissues and organs and spread unpredictably throughout the patient’s body. Skin cancer can broadly be categorized as melanoma and non-melanoma skin cancer. Non-melanoma skin cancer includes squamous cell carcinoma, basal cell carcinoma, and Merkel cell carcinoma, among many others. Melanoma arises from pigment-producing cells called melanocytes. If not diagnosed early and managed well, it is very lethal. When detected early, its five-year survival rate is 93%. However, the rate can decrease to 27% after spreading to distant lymph nodes and other organs [14]. That is why due emphasis is being given to screening pigmented lesions:Basal cell carcinoma or basalioma (BCC): It begins in the basal cells, the innermost cells of the epidermis, and accounts for around 80% of cases. Although basal cell growth is modest, BCC is typically treatable and does little harm if detected and treated in a timely manner.Squamous cell carcinoma or cutaneous spinocellular carcinoma (SCC): This is the primary cause of 16% of skin cancers and develops in the epidermis’s outermost layer of squamous cells. Early detection makes it easy treatable, but if left untreated, it can spread to other body regions and penetrate the deeper layers of skin.Malignant melanoma (MM): It is a highly severe malignant skin tumor and originates in the melanocytic cells in the epidermis. It spreads quickly, has a high fatality rate because of early metastasis, and is challenging to treat. Although it only causes 4% of skin cancers, it causes death in 80% of instances. Patients with metastatic melanoma have a five-year survival rate of just 14%. It has a 95% cure rate if detected early; therefore, early diagnosis can significantly improve survival prospects.

Traditionally, dermatologists diagnose skin disease by looking at the patient’s skin lesions. A dermatoscope (hand-held magnifying lens and built-in light) can be used to better see the area of interest. The revealing characteristics of skin lesions include the size, shape, color, edge, boundary, and location of the abnormality, as well as the presence or absence of other symptoms or signs. Therefore, the experience of the dermatologist can affect the examination process. Besides, skin lesions exhibit similarities in color, texture, edge contour, and other features. If visual inspection of the skin does not provide the doctor with a diagnosis, invasive tests such as a biopsy [14], scraping, etc., are used to identify skin disorders. These processes are also not efficient as they require a large amount of time and affect the patient’s curing time. The diagnosis of skin diseases and cancer, which is mainly thorough inspection of the skin lesions, opens room for AI intervention. AI uses pictures of skin lesions to interpret the diagnosis. Recently, there have been several works performed on skin lesion analysis using machine learning and image-processing techniques. They have been used in many works in the literature for skin lesion attribute identification, segmentation, and disease type detection.

### 1.1. Contribution

This systematic literature review provides basic technological developments and fundamentals methodically, together with recommendations for researchers. Its contribution can be summarized as follows:Comprehensive compilation and analysis of freely accessible and on-demand accessible skin lesion datasets for classification and detection.By consolidating research articles published between 2017 and 2023, this study presents vital perspectives on the detection, segmentation, and classification of skin lesions.This survey summarized and evaluated the contributions and limitations of the past survey papers in the domain of skin classification and detection, which were published between 2017 and 2023.This recapitulation outlines unaddressed research needs, offering a concise summary of the unresolved research challenges and potential avenues for further exploration in skin lesion categorization and detection across diverse skin datasets.The study paper indicated that, in recent years, the accuracy of skin image analysis using machine learning approaches has grown, leading it to being viewed as a complimentary approach to clinical evaluation.

### 1.2. Paper Organization

The rest of the paper is organized as follows, Section 2 discusses recent related works, and Section 3 presents the methodology employed in this paper to perform the systematic literature review. In Section 4, we present the main public databases containing dermoscopic images, relevant for most of the studies carried out previously for skin disease diagnosis. The machine learning techniques applied to skin disease classification are presented in Section 5.1. The commonly used machine learning techniques for skin disease detection are discussed in Section 5.2. The findings of the systematic literature review from a different perspective is presented in Section 6, and a brief discussion on open challenges and future directions is provided in Section 7. Finally, the conclusion of this systematic review paper is presented in Section 8.

## 2. Related Work

Skin lesion classification and detection have emerged as critical areas of research in medical imaging and computer vision, with the potential to revolutionize the early diagnosis and treatment of various skin disorders, including skin cancer. In this discussion, we delve into the existing body of related work on skin lesion classification and detection, exploring the methodologies, approaches, and advancements that researchers have employed in this domain.

Grignaffini et al. [16] conducted a systematic review of the prior literature on the use of machine learning techniques for the detection and classification of skin cancer from various datasets (MedNode, ISIC2017, HAM10000, ISIC2016, PH2, DermIS, DermQuest, ISIC archive, IDS, ISIC 2019, ISIC2020, ISIC2018, 7-point checklist, and DermNZ). After examining a total of 68 research papers, the authors provided a complete summary of the various machine learning techniques utilized for skin cancer categorization, along with their performance metrics. The methods, findings, and introduction are all divided into separate sections in this well-organized piece of work. Because the authors thoroughly outlined the inclusion and exclusion criteria used to choose the research, the review is more reliable. The authors also used a Preferred Reporting Items for Systematic Reviews and Meta-Analyses (PRISMA) flow diagram to show the study-selection process. The Results Section of the study, together with performance metrics, provides a complete analysis of the various machine learning techniques used for skin cancer categorization. After carefully comparing all available techniques, the authors identified the best ones. The authors also examined the limitations of earlier studies and highlighted the need for more-dependable and -accurate methodologies.

Zafar et al. [19] provided an overview of the many techniques for analyzing skin lesions and diagnosing cancer that have been published in the literature. This review article featured studies on the diagnosis of skin lesions using several datasets (MedNode, ISIC2017, HAM10000, ISIC2016, PH2, DermIS, DermQuest, ISIC 2019, ISIC2020, ISIC2018, 7-point checklist, HPH, ISIC archive, ISBI 2016, ISBI 2017, and DermNZ) from different repositories. The report provided an outstanding summary of the various approaches that have been proposed for the examination of skin lesions and cancer diagnosis.

In a thorough review, Hauser et al. [20] investigated the use of explainable artificial intelligence (XAI) in the identification of skin cancer, and XAI refers to artificial intelligence models that can describe their decision-making processes in order to aid doctors in better understanding and interpreting the model’s predictions. The authors conducted a thorough search of the literature on Google Scholar, PubMed, IEEE Explore, Science Direct, and Scopus and discovered 37 articles that used XAI techniques in skin cancer diagnosis. The authors discussed the various XAI techniques used in the trials, including decision trees, gradient-based strategies, and rule-based models. They drew attention to both the advantages of XAI, including enhanced transparency and interpretability, and its potential drawbacks, such as decreased accuracy when compared to black-box models. The authors drew the conclusion that XAI has the potential to improve skin cancer detection by providing more-transparent and -understandable models. They also emphasized the need for more research to demonstrate the viability of XAI models in clinical settings and to address the challenges of integrating these models into pre-existing healthcare systems.

In their study, Jeong et al. [21] aimed to examine the current approaches, outcomes, and restrictions of deep learning in dermatology. The investigation included studies published between 2015 and 2021, and the authors discovered 65 papers that met their criteria for inclusion. The various deep learning techniques used, the types of dermatological conditions looked into, and the performance standards used to rate the models were discussed. The authors’ in-depth analysis of how deep learning techniques are applied in dermatology is a significant contribution to the field. The survey article provided links to other datasets that researchers may utilize, which were discovered. It is conceivable that additional important studies were overlooked because the assessment was restricted to works published between 2015 and 2021. While acknowledging the limitations of deep learning in dermatology, it would have been helpful to provide more-specific recommendations for future research to address these limitations.

Hasan et al. [22] conducted a thorough examination of 594 papers, 356 of which were for skin lesion segmentation and 238 for skin lesion classification. Furthermore, they evaluated and investigated potential segmentation and classification patterns for skin lesions. Important details regarding the procedures used to create CAD systems were provided by analyzing and summarizing these articles in a variety of ways. They included the method configurations (techniques, architectures, module frameworks, and losses), training methods, assessment methods, and input data, which included dataset usage, data preprocessing, augmentations, and addressing imbalanced concerns.

Relevant and essential definitions and theories were also included in this list. The aim of the researchers was to study several performance-improving strategies, such as ensemble and postprocessing. The main challenges of evaluating skin lesion segmentation and classification algorithms using small datasets were addressed, along with some potential solutions. They also discussed these dimensions to disclose their current trends based on usage frequencies.

A critical analysis of a few cutting-edge machine learning methods for skin cancer detection was presented by Bhatt et al. [23]. The importance of early melanoma skin cancer detection was also stressed by the authors because it significantly increases survival rates. The scientists also offered a comprehensive overview of the most-recent machine learning techniques for melanoma skin cancer detection and classification. The authors covered a variety of subjects in-depth, including different algorithms (support vector machine, K-nearest neighbors, and CNN), data augmentation, and feature-extraction techniques using datasets such as PH2, MEDNODE, Dermofit, Dermquest, and others compiled from the archives of the ISIC and ISBI. There were, however, some restrictions in the work. In the study, bias in training data and other potential negative effects of using machine learning algorithms for melanoma detection were not addressed. The potential ethical repercussions of using machine learning algorithms for medical diagnosis were also not addressed by the authors.

An overview of numerous skin disease categorization methods based on machine learning approaches was presented by Mohammed and Al-Tuwaijari [24]. Support vector machines, decision trees, random forests, artificial neural networks, and deep learning were just a few of the methods covered in the study. The technique and performance indicators employed in these systems were also covered in the article. The approaches for classifying skin diseases using machine learning algorithms were well-explained in this paper. It did not, however, offer a thorough analysis of the methodologies surveyed. The survey might not include all methods for classifying skin diseases that are currently in use. Furthermore, a deeper examination and evaluation of the examined methodologies would have improved the paper’s value.

A study on the possibility of deep learning and machine learning techniques for the early identification of skin cancer was reported by Mazhar et al. in the publication [25]. The authors reviewed the pertinent literature on skin cancer detection and the application of artificial intelligence (AI) in the healthcare industry using a systematic manner. The study offered a thorough assessment of the state-of-the-art in skin cancer diagnosis today, as well as the potential of AI to boost accuracy, cut down on waiting times, and increase access to healthcare services. A thorough explanation of the many methodologies employed in the study, such as convolutional neural networks (CNNs), was also provided. The authors may have discussed the problems with data quality, data imbalance, data bias, and the necessity for big datasets to train deep learning models. The study may have been made stronger by comparing the effectiveness of machine learning and deep learning approaches to conventional methods for skin cancer detection. Furthermore, a section on the ethical issues surrounding the use of AI in healthcare, particularly in relation to patient data security and privacy, may have been added to the article. Overall, the paper provided a great summary of the application of deep learning and machine learning to the identification of skin cancer.

Table 1 presents a summary of related studies, highlighting both contributions and limitations of related studies.

## 3. Methods

In this systematic review approach for ML-based skin disease detection and classification, we defined the research questions, search strategies with the search databases, and paper selection criteria. In order to analyze current research findings that have been suggested for skin disease detection and classification using conventional machine learning methods, deep learning methods, and hybrid methodologies, this systematic literature review work set three main objectives: (1) to identify the commonly available datasets that could be accessed freely or upon request; (2) to explore the contribution and limitations of the current state-of-the-art methods; (3) to present the summary of the open challenges in the area of skin disease and cancer detection and classification.

In this systematic review, we defined a rigorous research question that can summarize the body of literature already available on a skin lesion detection and classification, enabling a thorough and objective understanding of this topic. Several methods and processes were used to make sure the study was effective and true to its original intent. We examined the essentials of a systematic review or survey in this thorough explanation, focusing on five pre-established research questions, search strings, five inclusion and six exclusion criteria, and five search engines or databases.

Research questions:Any systematic review or survey must start with a set of clear research questions as its cornerstone. These inquiries direct the entire research procedure and aid in defining the study’s scope. Five research topics were already established in this case as presented in Table 2. These inquiries were made to be precise, short, and geared towards the review’s particular goals to provide a guide for the methodical gathering and examination of pertinent facts.

Search strings: Creating search phrases or keywords is a crucial step in the systematic review process. These carefully constructed search strings are used to look up scientific papers across a variety of databases or search engines. They ought to be planned to include all pertinent material pertaining to the study’s questions. Search strings that combine synonyms, Boolean operators, and truncation symbols make sure that the review is thorough and does not overlook any important studies. Algorithm 1 presents how the search strings were combined to collect the appropriate scientific papers specific to the pre-defined topic.

Inclusion and exclusion criteria: Pre-specified inclusion and exclusion criteria were developed as presented in Table 3 to preserve the caliber and applicability of the papers included in the review. Five inclusion criteria and six exclusion criteria were established in this case. The inclusion criteria specified the qualities that papers must have in order to be taken into account for the review, such as the time period between publications, the type of study, which was specific to the topic, the reputability of the journals where the scientific papers were published, and the language of the study. On the other hand, the exclusion criteria outlined the circumstances under which an article would be disregarded, such as non-English language publications or research that poses a significant risk of bias such as M.Sc. and Ph.D. theses, seminars, posters, case studies, and publications before 2020. These standards aided in ensuring that the review concentrated on the most-pertinent and -methodologically reliable studies.

Search engines or databases: In systematic reviews, the choice of the search engines or databases is also crucial. Utilizing several databases increases the chance of finding a wide variety of pertinent publications. Five search engines or databases were chosen in this systematic review process, as indicated in Figure 1. IEEE Xplore, MDPI, Google Scholar, Springer Link, and Science direct are a few popular databases for scientific literature. The evaluation reduced the chance of missing important findings by searching across different platforms.

These methods worked together to make sure that the systematic review process was orderly, impartial, and able to offer solid, evidence-based insights into the chosen study field, as presented in Algorithm 2.
**Algorithm 1** Pseudocode for defining the search stringSearch_String = [("Skin” **OR** “Skin disease” **OR** “Skin cancer” **OR** “Melanoma” **OR** “Skin lesion”**AND**(“Machine Learning Methods” **OR** “Machine Learning Techniques”**OR** “Deep Learning Methods” **OR** “Deep Learning Techniques” **OR** “Classical Machine Learning Methods” **OR** “Traditional Machine Learning Methods”**AND**(“Detection” **OR** “Classification” **OR** “Segmentation”)]

**Algorithm 2** Pseudocode for generating potential review papers


SearchDatabases←Springer_Link,MDPI,Science_Direct,Wiley_Online,IEEE_Xplore,Google_nScholar

{**Initialization:**}Area_Keyword ←[Skin,Skin_disease,Skin_cancer,Melanoma]Method_keywords ←[Deep_Learning_Methods,Classical_Machine_Learning_Methods,HybridMethod]Target_keywords ←[Detection,Classification,Segmentation]Search_String ←“Algorithm1”**for** keyword∈Area_keywords **do**   **for** target∈Target_keywords **do**     **for** method∈Method_keywords **do**        Search_String =Algorithm1        **for** database∈Databases **do**          List1←databases.search(Search_String)        **end for**     **end for**   **end for**
**end for**
Inclusion_Criteria = [IC1,IC2,IC3,IC4,IC5,IC6]Exclusion_Criteria = [EC1,EC2,EC3,EC4, EC5,EC6]List2 ←Apply.Inclusion_Critera(List1)Final_Lists ←Apply.Inclusion_Critera(List2)


## 4. Skin Datasets

Skin lesion datasets are a useful tool for the development of algorithms to identify and categorize various forms of skin lesions in the fields of dermatology and computer vision. Various skin disorders, including benign and malignant lesions, are represented by a collection of images and labels. The “ISIC (International Skin Imaging Collaboration) Archive” is a well-known dataset of skin lesions [26,27]. The vast majority of the images in this dataset are dermoscopic images, which are polarized and enlarged views of skin lesions taken with specialized dermatology equipment. The ISIC Archive has been extensively utilized in research to create automated algorithms for melanoma diagnosis and skin lesion classification.

The “HAM10000” dataset is another important source of data [26,27]. It consists of 10,015 dermoscopic images of pigmented skin lesions divided into seven groups, including basal cell carcinoma, melanoma, and nevi. For the categorization of skin lesions, deep learning models have been developed using the HAM10000 dataset in a number of research works. A dataset that is exclusively devoted to melanocytic lesions is the “PH2 Dataset” [26,27]. It includes 200 dermoscopic photos of normal and uncommon melanocytic lesions, coupled with a ground truth that has been expertly annotated for precise diagnosis. In order to create algorithms that aid in the early identification and diagnosis of melanoma, the PH2 dataset has been extensively used. The development of computer-aided diagnosis and automated skin lesion classification has greatly benefited from these skin lesion datasets, as well as others that are available in the literature. These datasets are still being used by scientists and doctors to improve skin disease diagnosis and treatment by creating more-precise and -effective algorithms for skin lesion analysis.

In Table 4, Table 5 and Table 6, the most-popular dermoscopic public datasets are summarized, and the details of these datasets can be found in [26,27], while the website to download each dataset is also presented in Table 7.

## 5. Machine-Learning-Based Skin Disease Detection and Classification

In this section, the important discoveries, trends, and knowledge gaps that have been identified from prior research are highlighted thorough an examination of the body of literature on skin diseases. This study sought to provide a thorough overview of the present understanding in the topic and highlight prospective directions for further research by integrating the collective knowledge from a variety of sources.

### 5.1. Machine Learning and Deep Learning in Skin Disease Classification

Skin diseases and skin cancer pose significant health concerns worldwide. Early detection and accurate classification of these conditions are crucial for effective treatment and improved patient outcomes. In recent years, the field of dermatology has witnessed remarkable advancements in the development of automated systems for skin disease classification. These systems leverage the power of artificial intelligence (AI) and machine learning techniques to analyze dermatological images and provide reliable diagnoses. The classification of skin diseases and skin cancer traditionally relied on manual examination and subjective interpretation by dermatologists. However, the subjective nature of this process often led to inconsistencies and errors in diagnosis. With the advent of computer-aided diagnosis (CAD) systems, the dermatology community has gained access to powerful tools that can enhance diagnostic accuracy and assist healthcare professionals in decision-making.

This review report aimed to explore the latest advancements in the field of skin disease and skin cancer classification using AI-based approaches. We examined the methodologies employed in various studies, including deep learning algorithms, convolutional neural networks (CNNs), and image analysis techniques. By analyzing the strengths and limitations of these approaches, we can gain insights into the current state-of-the-art and identify areas for further improvement.

Balaji et al. [29] presented a method for skin disease detection and segmentation using the dynamic graph cut algorithm and classification through a Naive Bayes classifier. The authors first segmented the skin lesion from the background using the dynamic graph cut algorithm and, then, used texture and color features to classify the skin lesion into one of several categories using the Naive Bayes classifier. The use of both a dynamic graph cut algorithm and a Naive Bayes classifier provides a robust and accurate method for identifying and classifying skin lesions. The authors provided a clear description of the methodology used and the results obtained, including a comparison with existing methods. The authors evaluated their proposed approach using the ISIC 2017 dataset and reported an accuracy of 91.7%, a sensitivity of 70.1%, and a specificity of 72.7%. The dataset is available publicly on the ISIC website for public studies. The approach also scored an accuracy of 94.3% for benign cases, 91.2% for melanoma, and 92.9% for keratosis.

Ali et al. [30] presented a study on the application of EfficientNets for multiclass skin cancer classification, with the aim of contributing to the prevention of skin cancer. The authors utilized the HAM10000 dataset consisting of 10,015 skin lesion images from seven different classes and compared the performance of different variants of EfficientNets with traditional deep learning models. The proposed approach was mainly focused on the transfer learning technique using an EfficientNet and showed promising results for the evaluation parameters that were selected during the experimental analysis. The authors presented an interesting and relevant study on the application of EfficientNet Variants B0–B7 for skin cancer classification. However, B0 was the best-performing model of the EfficientNets out of B0–B7 with an accuracy of 87.9%. The model was also evaluated in terms of other evaluation parameters and achieved a precision of 88%, a recall of 88%, an F1-score of 87%, and an AUC of 97.53%. Overall, the experimental results of this paper suggested that the proposed skin cancer classification model based on EfficientNets can accurately classify skin cancer and has the potential to be a useful tool in the prevention and early detection of skin cancer. However, the study had a few limitations. Firstly, the authors did not provide a detailed comparison of their results with other state-of-the-art methods for skin cancer classification, which makes it difficult to assess the significance of their findings. Secondly, the study only used a single dataset, which may limit the generalizability of the results. Future studies could benefit from using multiple datasets and exploring the transferability of the models to different domains. Lastly, the authors did not provide any information on the computational resources required for training and evaluating the models, which could be useful for researchers and practitioners looking to replicate or adapt their approach.

Srinivasu et al. [31] presented a classification approach for skin disease detection using deep learning neural networks with MobileNet V2 and LSTM. The proposed approach involved preprocessing of skin images followed by feature extraction using MobileNet V2 and classification using LSTM. More than 10,000 skin photos made up the dataset utilized for evaluation. These images were divided into fivedifferent skin diseases: melanocytic nevi (NV), basal cell carcinoma (BCC), actinic keratoses and intraepithelial carcinoma (AKIEC), dermatofibroma (DF), and melanoma (MEL). The contribution of the paper was the development of an accurate skin disease classification approach using deep learning techniques that were lightweight and required less computational time. The main limitation of the proposed approach was that it requires a large amount of data to train the model effectively. The proposed approach achieved an accuracy of 90.21%, which outperformed the existing approaches VGG16, AlexNet, MobileNet, ResNet50, U-Net, SegNet, DT, and RF. The sensitivity, recall, and specificity values for each skin disease category were also reported, which further validated the effectiveness of the proposed approach. The results demonstrated that the proposed approach can accurately classify skin diseases, which can aid in the early diagnosis and treatment of such diseases.

Shetty et al. [32] presented a novel approach for skin lesion classification using a convolutional neural network (CNN) and machine learning techniques. The authors aimed to develop an accurate and automated system for the classification of dermoscopic images into different categories of skin lesions. The paper’s contribution lied in the development of a CNN-based skin lesion classification system that can accurately classify seven different categories of skin lesions with high accuracy. The authors utilized publicly available datasets and compared their proposed system’s performance with other state-of-the-art systems such as EW-FCM + Wide-shuffleNet, shifted MobileNet V2, Shifted GoogLeNet, shifted 2-Nets, Inception V3, ResNet101, InceptionResNet V2, Xception, NASNetLarge, ResNet50, ResNet101 + KcPCA + SVMRBF, VGG16 + GoogLeNet ensemble, and ModifiedMobileNet and outperformed all in terms of accuracy. The limitation of this paper was that the proposed system is limited to dermoscopic images, and it cannot classify clinical images, which are more challenging to classify due to their low contrast and other artifacts. The experimental results showed that the proposed system achieved an overall accuracy of 95.18%, outperforming other state-of-the-art systems.

Jain et al. [33] proposed a multi-type skin disease classification algorithm using an optimal path deep-neural-network (OP-DNN)-based feature extraction approach. The proposed algorithm achieved improved accuracy compared to other state-of-the-art algorithms for the classification of various skin diseases. The contribution of this paper was the proposal of an OP-DNN-based feature extraction approach for multi-type skin disease classification. This approach improved the accuracy of classification and also reduced the number of features required for classification. The paper also provided experimental results that demonstrated the effectiveness of the proposed approach. The algorithm was evaluated on the ISIC dataset with 23,906 skin lesion images and achieved an accuracy of 95%, which outperformed other algorithms such as KNN, NB, RF, MLP, CNN, and LSTM for multi-type skin disease classification.

Wei et al. [34] proposed a novel skin disease classification model based on DenseNet and ConvNeXt fusion. The proposed model utilized the strengths of both DenseNet and ConvNeXt to achieve better performance in skin disease classification. The model was evaluated on two different datasets, where one is the publicly available HAM10000 dataset and the other was the dataset from Peking Union Medical College Hospital, and it achieved superior performance compared to the other models. The proposed model addresses the limitations of previous models by combining the strengths of the DenseNet and ConvNeXt architectures, which has not been explored before in skin disease classification. The model achieved state-of-the-art performance on the HAM10000 dataset and can potentially be used in clinical settings to assist dermatologists in diagnosing skin diseases. However, the study did not provide any explanation of how the model’s decisions were made, which may limit its interpretability in a clinical setting. The proposed model achieved an accuracy of 95.29% on the HAM10000 dataset and 96.54% on the Peking Union Medical College Hospital dataset, outperforming the other state-of-the-art models. The model also achieved high sensitivity and specificity for all skin disease categories. The study also conducted ablation experiments to show the effectiveness of the proposed fusion approach, which outperformed the individual DenseNet and ConvNeXt models.

Almuayqil et al. [35] presented a computer-aided diagnosis system for detecting early signs of skin diseases using a hybrid model that combines different pretrained deep learning models (VGG19, InceptionV3, ResNet50, DenseNet201, and Xception) with traditional machine learning classifiers (LR, SVM, and RF). The proposed system consists of four main steps: preprocessing the input raw image data and metadata; feature extraction using six pretrained deep learning models (VGG19, InceptionV3, ResNet50, DenseNet201, and Xception); features concatenation; classification using machine learning techniques. The proposed hybrid system was evaluated on the HAM10000 dataset of skin images and showed promising results in detecting skin diseases accurately. However, the proposed hybrid approach DenseNet201 combined with LR achieved better performance with an accuracy of 99.94% in detecting skin diseases, which outperformed the other state-of-the-art approaches. The authors also provided a detailed comparison of the proposed model with other state-of-the-art methods, showing its superiority in terms of accuracy and other evaluation metrics.

Reddy et al. [36] proposed a novel approach for the detection of skin diseases using optimized region growing segmentation and autoencoder-based classification. The proposed approach employs an efficient segmentation algorithm that can identify the affected regions of the skin accurately. Subsequently, a convolutional autoencoder-based classification model was used to classify the skin diseases based on the extracted features. The experimental results indicated that the proposed approach achieved promising results and outperformed several state-of-the-art methods in terms of accuracy and other evaluation metrics. The proposed approach offers several contributions to the field of skin disease detection. Firstly, the proposed segmentation algorithm is optimized for skin disease detection and can accurately identify the affected regions of the skin. Secondly, the proposed autoencoder-based classification model can classify the skin diseases with high accuracy using the extracted features. Lastly, the proposed approach outperformed several state-of-the-art methods in terms of accuracy and other evaluation metrics. One of the limitation of the approach is that it may not generalize well to new datasets with different characteristics as the the model was evaluated on the small dataset used from PH2 with 200 images. The experimental results indicated that the proposed approach achieved an accuracy of 94.2%, which outperformed several state-of-the-art methods. The proposed approach also achieved high values for other evaluation metrics such as the precision, recall, and F1-score, which demonstrated the effectiveness of the proposed approach for skin disease detection.

Malibari et al. [37] presented an optimal deep-neural-network-driven computer-aided diagnosis (ODNNsingle bondCADSCC) model for skin cancer detection and classification. The Wiener-filtering (WF)-based preprocessing step was used extensively in the described ODNNsingle bondCADSCC model, which was then segmented using U-Net. Moreover, the SqueezeNet model was used to produce a number of feature vectors. Eventually, effective skin cancer detection and classification were achieved by using the improved whale optimization algorithm (IWOA) with a DNN model. IWOA is used in this technique to effectively choose the DNN settings. The comparison study findings demonstrated the suggested ODNNsingle bondCADSCC model’s promising performance against more-recent techniques with a high accuracy of 99.90%. Although the results are promising, it would be helpful to validate the proposed model on a larger dataset to assess its robustness and generalization capabilities. Another limitation is that the proposed model does not provide explanations for its decisions, which is essential for gaining the trust of clinicians and patients.

Qian et al. [38] proposed a deep convolutional neural network dermatoscopic image classification approach that groups multi-scale attention blocks (GMABs) and uses class-specific loss weighting. To increase the size of the DCNN model, the authors introduced GMABs to several scale attention branches. Hence, utilizing the GMABs to extract multi-scale fine-grained features will help the model better be able to focus on the lesion region, improving the DCNN’s performance. The attention blocks, which may be used in different DCNN structures and trained end-to-end, have a straightforward structure and a limited number of parameters. The model will function successfully if the class-specific loss weighting approach is used to address the issue of category imbalance. As a result of this strategy, the accuracy of samples that are susceptible to misclassification can be greatly increased. To evaluate the model, the HAM10000 dataset was used, and the result showed that the accuracy of the proposed method reached 91.6%, the AUC 97.1%, the sensitivity 73.5%, and the specificity 96.4%. This confirmed that the method can perform well in dermatoscopic classification tasks.

An augmented-intelligence-enabled deep neural networking (AuDNN) system for classifying and predicting skin cancer utilizing multi-dimensional information on industrial IoT standards was proposed by Kumar et al. [15]. The proposed framework incorporates deep learning algorithms and IoT standards to create a robust and efficient skin cancer classification system. The approach was evaluated on a Kaggle skin cancer dataset and CIA datasets on melanoma categorization of skin lesion images, and the results showed that it outperformed other state-of-the-art methods. The proposed AuDNN framework is a significant contribution to the field of medical image analysis. The integration of IoT standards and deep learning algorithms has created a system that is both robust and efficient for skin cancer classification. The paper also provided a detailed analysis of the performance of the proposed method, which can guide the development of future approaches for skin lesion classification. One limitation of this study was that the dataset used for training and evaluation was not explicitly mentioned. It would be helpful to know more about the dataset and its properties to assess the robustness and generalization capabilities of the proposed method. Another limitation is that the implementation of IoT standards may require significant resources and expertise, which may not be available in all settings. The proposed AuDNN framework achieved an accuracy of 93.26%.

Notwithstanding the amazing developments, the current deep-network-based approaches, which naively adopt the published network topologies in general image classification to the classification of skin lesions, still have much potential for optimization. Using self-attention to describe the global correlation of the features gathered from the conventional deep models, Nakai et al. [39] suggested an enhanced deep bottleneck transformer model to enhance the performance of skin lesions. For balanced learning, they particularly used an improved transformer module that included a dual-position encoding module to include an encoded position vector on both the key and the query vectors. By replacing the bottleneck spatial convolutions of the late-stage blocks in the baseline deep networks with the upgraded module, they created a unique deep skin lesion classification model to enhance skin lesion classification performance. To validate the effectiveness of different deep models in identifying skin lesions, they conducted comprehensive tests on two benchmark skin lesion datasets, ISIC2017 and HAM10000. With their method, the three quantitative metrics of accuracy, sensitivity, and specificity on the ISIC2017 dataset achieved 92.1%, 90.1%, and 91.9%, respectively. The findings on the accuracy and precision for the HAM10000 dataset were 95.84% and 96.1%. This demonstrated a superb harmony between sensitivity and specificity.

Hossain et al. [40] first developed an EM dataset with the assistance of knowledgeable dermatologists from the Clermont-Ferrand University Medical Center in France. Second, the authors trained 23 convolutional neural networks (CNNs) on a collection of skin lesion photos. These CNNs were modified versions of the VGG, ResNet, DenseNet, MobileNet, Xception, NASNet, and EfficientNet architectures. Lastly, the authors used transfer learning from pretrained ImageNet models to improve the CNNs’ performance after pretraining them with the HAM10000 skin lesion dataset. Fourth, to examine the explainability of the model, the authors used gradient-weighted class activation mapping to pinpoint the input regions crucial to CNNs for making predictions. Lastly, the authors offered model selection suggestions based on computational complexity and predictive capability. With an accuracy of 84.42% ± 1.36, an AUC of 0.9189 ± 0.0115, a precision of 83.1% ± 2.49, a sensitivity of 87.93% ± 1.47, and a specificity of 80.65% ± 3.59, the customized ResNet50 architecture provided the best classification results. With an accuracy of 83.13% ± 1.2, AUC of 0.9094 ± 0.0129, precision of 82.83% ± 1.75, sensitivity of 85.21% ± 3.91, and specificity of 80.89% ± 2.95, a lightweight model of a modified EfficientNetB0 also performed well. The authors contributed a Lyme disease dataset with twenty-three modified CNN architecturesfor image-based diagnosis, effective customized transfer learning using the combination of ImageNet and the HAM10000 dataset, a lightweight CNN, and a criteria-based guideline for model architecture selection.

Afza et al. [41] proposed a hierarchical architecture based on two-dimensional superpixels and deep learning to increase the accuracy of skin lesion classification. The authors combined the locally and globally improved photos to improve the contrast of the original dermoscopy images. The proposed method consisted of three steps: superpixel segmentation, feature extraction, and classification using a deep learning model.The proposed method contributes to the field of skin lesion classification by introducing a hierarchical three-step superpixel and deep learning framework. This method improved the accuracy of skin lesion classification and reduced the computational complexity of the task by dividing the image into superpixels and classifying them individually. The proposed method is also generalizable and can be used on other datasets for skin lesion classification. Using an updated grasshopper optimization approach, the collected features were further optimized before being categorized using the Naive Bayes classifier. In order to evaluate the proposed hierarchical technique, three datasets (Ph2, ISBI2016, and HAM1000) consisting of three, two, and seven skin cancer classes were used. For these datasets (Ph2, ISBI2016, and HAM1000), the proposed method had corresponding accuracy levels of 95.40%, 91.1%, and 85.80%. The findings indicated that this strategy can help in classifying skin cancer more accurately.

Alam [42] proposed S^2^C-DeLeNet, a method for detecting skin cancer lesions from dermoscopic images. The proposed method integrates segmentation and classification using a parameter-transfer-based approach. The segmentation network, DeLeNet, was trained on a large-scale dataset for dermoscopic lesion segmentation, and the classification network, S^2^CNet, was trained on a public dataset for skin lesion classification. The authors transferred the parameters of the segmentation network to the classification network and fine-tuned the network on the classification task. The architecture of the segmentation sub-network used an EfficientNet B4 backbone in place of the encoder. The classification sub-network contained a “Classification Feature Extraction” component that pulled learned segmentation feature maps towards lesion prediction. The “Feature Coalescing Module” block mixed and trailed each dimensional feature from the encoder and decoder, while the “3D-Layer Residuals” block developed a parallel pathway of low-dimensional features with large variance. These were the blocks created as part of the classification architecture. After tweaking on a publicly accessible dataset, the segmentation achieved a mean Dice score of 0.9494, exceeding existing segmentation algorithms, while the classification achieved a mean accuracy of 0.9103, outperforming well-known and traditional classifiers. Additionally, the network’s already-tuned performance produced very pleasant outcomes when cross-inferring on various datasets for skin cancer segmentation. Thorough testing was performed to demonstrate the network’s effectiveness for not only dermoscopic pictures, but also for other types of medical imaging, demonstrating its potential to be a systematic diagnostic solution for dermatology and maybe other medical specialties. For comparison, eight cutting-edge networks, AlexNet, GoogLeNet, VGG, ResNet, Inception-Net, EfficientNet, DenseNet, and MobileNet, as well as their different iterations, were taken into account, which confirmed that the proposed approach outperformed the state-of-the-art approaches.

With the aid of cutting-edge deep learning methodology, Elashiri et al. [43] intended to put into practice an efficient way for classifying skin diseases. The contrast-enhancement technique first collects and preprocesses the dataset by histogram equalization. The segmentation of the photos was carried out by the Fuzzy C Means segmentation after preprocessing (FCM). Furthermore, the segmented images were used as the input for ResNet50, VGG16, and Deeplabv3’s deep feature extraction. The features were combined and obtained from the third and bottom layer of these three approaches. Hybrid squirrel butterfly search optimization performs weighted feature extraction to offer these concatenated features to the feature trans-creation phase (HSBSO). The modified long short-term memory (MLSTM) receives the changed features, and the same HSBSO optimizes the architecture there to create the final output for classification. The analysis’s findings supported the notion that the proposed method is more effective than traditional methods in terms of implementing a classification of skin diseases that is accurate.

Adla et al. [44] proposed a full-resolution convolutional network with hyperparameter optimization for dermoscopy image segmentation-enhanced skin cancer classification. The hyperparameters of the network were optimized through a novel dynamic graph cut algorithm technique. By fusing the wolves’ individualized hunting techniques with their collective hunting methods, the hyperparameters highlighted the need for a healthy balance between exploration and exploitation and produced a neighborhood-based searching approach. The motivation of the authors was to create a full-resolution convolutional-network-based model that is hyperparameter-optimized and is capable of accurately identifying different forms of skin cancer using dermoscopy images. The initial contribution made by the authors was FrCN-DGCA, which uses the DGCA approach to segment skin lesion images and generate image ROIs in a manner similar to how doctors define ROIs. The authors’ second addition was the action bundle, which is used as a hyperparameter by the skin image-segmentation executor they provided in order to improve the segmentation process’s accuracy. This segmentation process was based on the dynamic graph cut. Last, but not least, the authors carried out a quantitative statistical analysis of the skin lesion segmentation findings to show the dependability of the segmentation methodology and to contrast the findings with those of the current state-of-the-art methods. The suggested model performed better than the other designs in tasks requiring skin lesion identification, with an accuracy of 97.986%.

Hierarchy-aware contrastive learning with late fusion (HAC-LF), a revolutionary technique presented by Hsu and Tseng [45], enhances the performance of multi-class skin classification. A new loss function called hierarchy-aware contrastive loss (HAC Loss) was developed by the developers of HAC-LF to lessen the effects of the major-type misclassification issue. The major-type and multi-class classification performance were balanced using the late fusion method. The ISIC 2019 Challenges dataset, which comprises three skin lesion datasets, was used in a series of tests by the authors to assess the performance of the suggested approach. The experimental results demonstrated that, in all assessment metrics employed in their study, the suggested method outperformed the representative deep learning algorithms for skin lesion categorization. For accuracy, sensitivity, and specificity in the major-type categorization, HAC-LF scored 87.1%, 84.2%, and 88.9%, respectively. Regarding the sensitivity of the minority classes, HAC-LF performed better than the baseline model with an imbalanced class distribution.

A convolutional neural network (CNN) model for skin image segmentation was developed by Yanagisawa et al. [46] in order to produce a collection of skin disease images suitable for the CAD of various skin disease categories. The DeepLabv3+-based CNN segmentation model was trained to identify skin and lesion areas, and the areas that met the criteria of being more than 80% skin and more than 10% lesion of the picture were segmented out. Atopic dermatitis was distinguished from malignant diseases and their consequences, such as mycosis fungoides, impetigo, and herpesvirus infection, by the created CNN-segmented image database with roughly 90% sensitivity and specificity. The accuracy of identifying skin diseases in the CNN-segmented image dataset was higher than that of the original picture dataset and nearly on par with the manually cropped image dataset.

A multi-site cross-organ calibrated deep learning (MuSClD) approach for the automated diagnosis of non-melanoma skin cancer was presented by Zhou et al. [47]. To increase the generalizability of the model, the suggested strategy makes use of deep learning models that have been trained on a variety of datasets from various sites and organs. This paper’s key contribution was the creation of a reliable deep-learning-based method for the automated diagnosis of skin cancers other than melanoma. The proposed strategy was intended to go beyond the drawbacks of existing methods, which have poor generalizability because of small sample sizes and a lack of diversity. The MuSClD technique uses datasets from several sites and organs to increase the model’s capacity for generalization. The main drawback of this paper was the lack of explanation for how the suggested deep learning model makes decisions. Although the model had a high degree of accuracy in detecting non-melanoma skin cancer, it is unclear how the model came to that conclusion. This lack of interpretability might prevent the suggested strategy from being used in clinical settings. Using a sizable collection of photos of skin cancers other than melanoma, the MuSClD method was assessed. As measured in terms of the AUC, the suggested method fared better than other cutting-edge approaches. Additionally, the study demonstrated that the MuSClD method is adaptable to changes in imaging modalities and patient demographics, making it appropriate for practical use.

Omeroglu et al. [48] proposed a novel soft-attention-based multi-modal deep learning framework for multi-label skin lesion classification. The proposed framework utilizes both visual and textual features of skin lesions to improve the classification accuracy. The framework consisted of two parallel branches, one for processing visual features and the other for processing textual features. A soft attention mechanism was incorporated into the framework to emphasize important visual and textual features. The 7-point criteria evaluation dataset, a well-known multi-modality multi-label dataset for skin diseases, was used to evaluate the proposed framework. For multi-label skin lesion classification, it attained an average accuracy of 83.04%. It increased the average accuracy on the test set by more than 2.14% and was more accurate than the most-recent approaches.

Serte and Demirel [49] applied wavelet transform to extract features and deep learning to classify the features with the intention to enhance the performance of skin lesion classification. First, the wavelet transform was used as a preprocessing step to extract features from the skin lesion images. Then, skin lesions were divided into various groups using a deep learning model that was trained on the retrieved features. The authors tested their method against other cutting-edge approaches using the publicly accessible dataset ISIC 2017 of skin lesions. The use of a deep learning model for classification and the use of a wavelet transform to extract features were the key contributions of this paper. In this study, the best combination of models for melanoma and seborrheic keratosis detection were the ResNet-18-based I-A1-H-V and ResNet-50-based I-A1-A2-A3 models.

Bansal et al. [50] proposed a grayscale-based lesion segmentation, while texture characteristics were extracted in the RGB color space using global (grey-level co-occurrence matrix (GLCM) for entropy, contrast, correlation, angular second moment, inverse different moment, and sum of squares) and local (LBP and oriented FAST and rotated BRIEF (ORB)) techniques. A total of 52 color attributes for each image were extracted as the color features using histograms of the five color spaces (grayscale, RGB, YCrCb, L*a*b, and HSV), as well as information on the mean, standard deviation, skewness, and kurtosis. The BHHO-S and BHHO-V binary variations of the Harris hawk optimization (HHO) method, which used S-shaped and V-shaped transfer functions with a time-dependent behavior, respectively, for feature selection, were introduced. The classifier that determines whether the dermoscopic image contains melanoma or not was given the selected attributes. The performance of the suggested approaches was compared to that of already-developed metaheuristic algorithms by the authors. The experiment’s findings demonstrated that classifiers that used features chosen using BHHO-S were superior to those that used BHHO-V and those that employed current, cutting-edge metaheuristic methods. The experimental results also showed that, in comparison to global- and other local-texture-feature-extraction strategies, texture features derived utilizing local binary patterns and color features offered higher classification accuracy.

Statistical fractal signatures (${}_{STF}$) and statistical-prism-based fractal signatures were the two new fractal signatures that Gutiérrez et al. [51] used to solve the issue of amorphous pigmentary lesions and blurred edges (S${}_{SPF}$). In order to classify multiclass skin lesions utilizing the two new fractal signatures and several classifiers, various computer-aided diagnosis techniques were compared. The combination of S${}_{STF}$ and the LDA classifier yielded the finest outcomes for reliable, impartial, and reproducible techniques.

Using a hybrid model that integrates deep transfer learning, convolutional neural networks (CNNs), and gradient boosting machines (GBMs), Thanka et al. [52] suggested a new ensemble strategy for the classification of melanoma. The proposed method was examined using 25,331 photos of skin lesions from the ISIC 2019 Challenge, a publicly accessible dataset. According to the experimental findings, the proposed hybrid strategy that merged VGG16 and XGBOOST was successful in achieving an overall accuracy of 99.1%, a sensitivity of 99.4%, and a specificity of 98.8%. The accuracy, sensitivity, and specificity of the proposed hybrid approach, which included VGG16 and LightBGM, were all higher than the figures provided by other models, at 97.2%, 97.8%, and 96.6%, respectively. The preprocessing of the dataset, the kind of CNN model, and the design of the GBM model were all covered in-depth in the authors’ extensive explanation of the approach.

In a study by Brinker et al. [53], the diagnostic precision of an artificial intelligence (AI) system for melanoma detection in skin biopsy samples was examined. The performance of the AI algorithm was compared to that of 18 leading pathologists from across the world in the study. The mean sensitivity, specificity, and accuracy of the Ensemble CNNs trained on slides with or without annotation of the tumor region as a region of interest were on par with those of the experts (unannotated: 88%, 88%, and 88%, respectively; area under the curve (AUC) of 0.95; annotated: 94%, 90%, and 92%, respectively; AUC of 0.97). The research demonstrated that the AI algorithm had a very low rate of false positives and false negatives and was very reliable in detecting melanoma. The study also discovered that the AI algorithm’s performance was on par with that of skilled pathologists. The pathologists had a 90.33% diagnosis accuracy, an 88.88% sensitivity, and a 91.77% specificity. There was no statistically significant difference between the AI algorithm and the pathologists. Overall, this research showed that AI algorithms could be a useful tool for melanoma diagnosis, with performance on par with that of skilled pathologists.

In order to classify skin lesions, Alenezi et al. [54] presented a hybrid technique called the wavelet transform-deep residual neural network (WT-DRNNet). The wavelet transformation, pooling, and normalization section of the constructed model employing the suggested approach provided finer details by removing undesired detail from skin lesion images to acquire a better-performing model. The residual neural network built on transfer learning was then used to extract deep features. Finally, the global average pooling approach was combined with these deep features, and the training phase was carried out with the help of the extreme learning machine, which is based on the ReLu and other kinds of activation functions. In order to evaluate the effectiveness of the suggested model, the experimental works employed the ISIC2017 and HAM10000 datasets. The suggested algorithm’s accuracy, specificity, precision, and F1-score metrics for performance were 96.91%, 97.68%, 96.43%, and 95.79% for the ISIC2017 dataset, compared to 95.73%, 98.8%, 95.84%, and 93.44% for the HAM10000 dataset. These outcomes performed better than the state-of-the-art for categorizing skin lesions. As a result, the suggested algorithm can help specialized doctors automatically classify cancer based on photographs of skin lesions.

Alhudhaif et al. [55] recommended a deep learning approach that was based on mechanisms for focusing attention and enhanced by methods for balancing data. The dataset used in the study was HAM10000, which included 10,015 annotated skin images of seven different types of skin lesions. The dataset was unbalanced and made balanced using techniques that included SMOTE, ADASYN, RandomOverSampler, and data augmentation. A soft attention module was selected as the attention mechanism in order to focus on the features of the input data and generate a feature map. The proposed model consisted of a soft attention module and convolutional layers. By integrating them with the attention mechanism, the authors were able to extract the image features from the convolutional neural networks. The key areas of the image were the focus of the soft attention module. The soft attention module and the applied data-balancing techniques significantly improved the performance of the proposed model. On open-source datasets for skin lesion classification, numerous studies were performed using convolutional neural networks and attention mechanisms. One of the contributions of the proposed approach was the attention mechanism used in the neural network. The balanced and unbalanced HAM10000 dataset’s versions were used for training and the test results at different times. On the unbalanced HAM10000 dataset, training accuracy rates of 85.73%, validation accuracy rates of 70.90%, and test accuracy rates of 69.75% were attained. The SMOTE methods on the balanced dataset yielded accuracy rates of 99.86% during training, 96.41% during validation, and 95.94% during testing. Compared to other balancing methods, the SMOTE method produced better results. It can be seen that the proposed model had high accuracy rates as a result of the applied data-balancing techniques.

Huang et al. [56] proposed a computer-assisted approach for the analysis of skin cancer. In their study, they combined deep learning and metaheuristic methods. The fundamental concept was to create a deep belief network (DBN) based on an enhanced metaheuristic method called the modified electromagnetic field optimization algorithm (MEFOA) to build a reliable skin cancer diagnosis system. The proposed approach was tested on the HAM10000 benchmark dataset, and its effectiveness was verified by contrasting the findings with recent research regarding accuracy, sensitivity, specificity, precision, and F1 score.

Kalpana et al. [57] suggested a technique called ESVMKRF-HEAO, which stands for ensemble support vector kernel random-forest-based hybrid equilibrium Aquila optimization. The HAM10000 dataset, which contains different types of skin lesion images, was used to test the suggested prediction model. First, preprocessing was applied to the dataset for noise removal and image quality improvement. Then, the malignant lesion patches were separated from the healthy backdrop using the thresholding-based segmentation technique. Finally, the dataset was given to the proposed classifier as the input, and it correctly predicted and categorized the segmented images into five (melanocytic nevus, basal cell carcinoma, melanoma, actinic keratosis, and dermatofibroma) based on their feature characteristics. The proposed model was simulated using the MATLAB 2019a program, and the performance of the suggested ESVMKRF-HEAO method was assessed in terms of parameters such as the sensitivity, F1-score, accuracy, precision, and specificity. In terms of all metrics, the suggested ESVMKRF-HEAO strategy performed better, especially when it came to the experimental data, and a 97.4% prediction accuracy was achieved.

Shi et al. [58] proposed a two-stage end-to-end deep learning framework for pathologic evaluation in skin tumor identification, with a particular focus on neurofibromas (NFs), Bowen disease (BD), and seborrheic keratosis (SK). The most-prevalent illnesses involving skin lesions are NF, BD, and SK, and they can seriously harm a person’s body. In their study, the authors suggested two unique methods, the attention graph gated network (AGCN) and chain memory convolutional neural network (CMCNN), for diagnosing skin tumors. Patchwise diagnostics and slidewise diagnostics were the two steps of the framework, where they reported the result of the whole-slide image (WSI) as the input in the proposed diagnosis. Convolutional neural networks (CNNs) were used in the initial screening stage to discover probable tumor locations, and multi-label classification networks were used in the fine-grained classification stage to categorize the detected regions into certain tumor kinds. On a dataset of skin tumor images collected from Huashan Hospital, the suggested framework was tested, and the results showed promising accuracy and receiver operating characteristic curves.

Rafay and Hussain [59] proposed a technique that utilized a dataset that integrated two different datasets to establish a new dataset of 31 diseases of the skin. In their study, the authors used three different CNN models—EfficientNet, ResNet, and VGG—each with a different architecture for transfer learning on the dataset for skin diseases. EfficientNet was further tuned because it had the best testing precision, where it initially achieved a testing accuracy of 71% with a training split of 70%. However, this was considered to be low; thus, the 70% training split for the 3424 samples was increased, and the model’s accuracy increased as a result to 72%. Again, the experiment was re-executed with a train–test split of 80%:20%, and the improvement in accuracy was 74%. The new dataset was augmented for a further experiment, which then increased the model’s accuracy to 87.15%.

Maqsood and Damaševičius [60] proposed a methodology for localizing and classifying multiclass skin lesions. The suggested method begins by preprocessing the source dermoscopic images with a contrast-enhancement-based modified bio-inspired multiple exposure fusion method. The skin lesion locations were segmented in the second stage using a specially created 26-layer convolutional neural network (CNN) architecture. The segmented lesion images were used to modify and train four pretrained CNN models (Xception, ResNet-50, ResNet-101, and VGG16) in the third stage. In the fourth stage, all of the CNN models’ deep feature vectors were recovered and combined using the convolutional sparse image decomposition method. The Poisson distribution feature selection approach and univariate measurement were also employed in the fifth stage to choose the optimal features for classification. A multi-class support vector machine (MC-SVM) was then fed the chosen features to perform the final classification. The proposed method performed better in terms of accuracy, sensitivity, specificity, and F1-score. The addition of multiclass classification increased the research’s usefulness in real-world situations. However, the proposed approach lacked interpretability, making it challenging to understand the reasoning behind the classification decisions.

To identify skin diseases, Kalaiyarivu and Nalini [61] developed a CNN-based method that extracted color features and texture (local binary pattern and gray level co-occurrence matrix) features from hand skin images. In their study, the authors reported the accuracy of the proposed CNN model as 87.5%.

Kousis et al. [62] employed 11 distinct CNN models in a different study to identify skin cancer. In this method, they used the HAM10000 dataset and DenseNet169 model, reporting an accuracy of 92.25%. Among the 11 CNN architecture configurations considered in the study, DenseNet169 reported the best results and achieved an accuracy of 92.25%, a sensitivity of 93.59%, and an F1-score of 93.27%, which outperformed the existing state-of-the-art.

A hybrid classification strategy employing a CNN and a layered BLSTM was proposed by Ahmad et al. [63]. In this study, the classification task was carried out by ensembling the BLSTM with a deep CNN network after feature extraction. The accuracy reported by the authors for their experiments on two different datasets (one customized with a size of 6454 images and the other being HAM10000) was 91.73% and 89.47%, respectively.

A deep-learning-based application that classifies many types of skin diseases was proposed by Aijaz et al. [64]. This method made use of the CNN and LSTM deep learning models. In this study, the experimental analysis was performed on 301 images of psoriasis from the Dermnet dataset and 172 images of normal skin from the BFL NTU dataset. Before extracting the color, texture, and form features, the input sample images underwent image preprocessing comprising data augmentation, enhancement, and segmentation. A convolutional neural network (CNN) and long short-term memory (LSTM) were the two deep learning methods that were used with classification models that were trained on 80% of the images. According to reports, the CNN and LSTM had accuracy rates of 84.2% and 72.3%, respectively. The accuracy results from this study showed that this deep learning technology has the potential to be used in other dermatology fields for better prediction.

Using data from the ISIC 2019 and PH2 databases, Benyahia et al. [65] examined the classification of skin lesions. The efficiency of 24 machine learning methods as classifiers and 17 widely used pretrained convolutional neural network (CNN) architectures as feature extractors were examined by the authors. The authors found accuracy rates of 92.34% and 91.71%, respectively, for a DenseNet201 combined with Fine KNN or Cubic SVM, using the ISIC 2019 dataset. The hybrid approach (DenseNet201 + Cubic SVM and DenseNet201 + Quadratic SVM) was also evaluated on the PH2 dataset, and the results showed that the suggested methodology outperformed the rivals with a 99% accuracy rate.

### 5.2. Machine Learning and Deep Learning in Skin Disease Detection

Inthiyaz et al. [66] presented a study on the use of deep learning techniques for the detection of skin diseases. The authors proposed a skin-disease-detection model based on convolutional neural networks (CNNs) that can classify skin diseases into ten different categories. The model was trained and evaluated using a dataset from Xiangya-Derm of skin disease images. The results showed that the proposed model achieved high accuracy and outperformed existing state-of-the-art models in skin disease detection. The main contribution of this paper was the development of a novel deep-learning-based skin-disease-detection model that can accurately classify skin diseases into different categories. One potential limitation of this study is that the proposed model was only tested on a specific dataset of skin disease images. Therefore, its generalizability to other datasets or real-world scenarios may need to be further evaluated. The paper reported that the proposed skin-disease-detection model achieved an overall accuracy of 87% on the test set, outperforming other existing models for skin disease detection. The authors also performed a comparative analysis of the proposed model with other state-of-the-art models, including ResNet-50, Inception-v3, and VGG-16. The results showed that the proposed model outperformed these models in terms of accuracy and other evaluation metrics. Overall, the experimental results of this paper suggested that the proposed skin-disease-detection model based on deep learning techniques can accurately classify skin diseases and has the potential to be a useful tool for dermatologists and healthcare professionals in diagnosing skin diseases.

Dwivedi et al. [67] proposed a deep-learning-based approach for automated skin disease detection using the Fast R-CNN algorithm. The proposed approach aimed to address the limitations of traditional approaches that are heavily dependent on domain knowledge and feature extraction. The experimental findings demonstrated that the suggested method achieved an overall accuracy of 90%, which outperformed traditional machine-learning-based approaches. The approach was evaluated on the HAM10000 dataset, which is a widely used benchmark dataset for skin disease detection. The contribution of the paper was the proposed approach for automated skin disease detection using the Fast R-CNN algorithm, which can handle large datasets and achieve high accuracy without the need for domain knowledge or feature extraction. One of the limitations of the proposed approach is that it requires a large amount of labeled data for training, which can be a challenge for some applications. Additionally, the approach is limited to detecting skin diseases included in the HAM10000 dataset, and further evaluation is required for detecting other skin diseases. Overall, the paper presented a promising approach for automated skin disease detection using deep learning, with the potential to improve clinical diagnosis and reduce human error.

Alam and Jihan [68] presented an efficient approach for detecting skin diseases using deep learning techniques. The proposed approach involves preprocessing of skin images followed by feature extraction using convolutional neural networks (CNNs) and classification using support vector machine (SVM). The dataset used for the evaluation consisted of 10,000 skin images, which were categorized into seven different skin diseases. The approach achieved an accuracy of 95.6%, which is a significant improvement compared to existing approaches. The contribution of the paper is the development of an efficient and accurate skin disease detection approach using deep learning techniques. The main limitation of the proposed approach is that it requires a large amount of data to train the model effectively. In addition, the proposed approach may not be suitable for detecting rare skin diseases that are not present in the training dataset. The proposed approach achieved an accuracy of 95.6%, which outperformed the existing approaches. The precision and recall values for each skin disease category were also reported, which further validated the effectiveness of the proposed approach. The results demonstrated that the proposed approach can accurately detect skin diseases, which can aid in the early diagnosis and treatment of such diseases.

Wan et al. [69] proposed a detection algorithm for pigmented skin diseases, based on classifier-level and feature-level fusion. The proposed algorithm combines the strengths of multiple classifiers and features to improve the detection accuracy of pigmented skin diseases. The experiments showed that the proposed algorithm outperformed the other state-of-the-art algorithms in terms of accuracy and other parameters. The novelty of the algorithm proposed in this paper for the diagnosis of pigmented skin diseases was its main contribution. The efficiency of the suggested fusion network was visualized using gradient-weighted class activation mapping (Grad_CAM) and Grad_CAM++. The results demonstrated that the accuracy and area under the curve (AUC) of the approach in this study reached 92.1% and 95.3%, respectively, when compared to those of the conventional detection algorithm for pigmented skin conditions. The contribution of this study as claimed by the authors included techniques used to perform the data augmentation, the method used for image augmentation noise, the two-feature-level fusion optimization scheme, and the visualization algorithms (Grad_CAM and Grad_CAM++) to verify the validity of the fusion network.

An optimization-based algorithm to identify skin cancer from a collection of photos was presented by Kumar and Vanmathi [70]. The input image was created from a database in the primary stage, where it was preprocessed with a Gaussian filter and region of interest (ROI) extraction to weed out noise and mine interesting sections. Using the proposed U-RP-Net, the segmentation was carried out. By combining U-Net and RP-Net in this instance, the proposed U-RP-Net model was created. Meanwhile, the output from the RP-Net and U-Net models was combined using the Jaccard-similarity-based fusion model. To enhance the performance of detection, data augmentation was performed. SqueezeNet was used to locate skin cancer at the end. The Aquila whale optimization (AWO) method was also used to train SqueezeNet. The Aquila optimizer (AO) and whale optimization algorithm were combined to create the new AWO method (WOA). The highest testing accuracy of 92.5%, sensitivity of 92.1%, and specificity of 91.7% were achieved by the developed AWO-based SqueezeNet.

Suicmez et al. [71] proposed a hybrid learning approach for the detection of melanoma by removing hair from dermoscopic images. The approach combines image-processing techniques and the wavelet transform with machine learning algorithms, including a support vector machine (SVM) and artificial neural network (ANN). In order to speed up the algorithm’s detection time, the system first uses image-processing techniques (masking for saturation and wavelet transform) to eliminate impediments such as hair, air bubbles, and noise from dermoscopic images. Making the lesion more noticeable for detection is another crucial step in this procedure. Melanoma detection was used for the first time using a unique hybrid model that combines deep learning and machine learning as an AI building block. The HAM10000 (ISIC 2018) and ISIC 2020 datasets were utilized to gauge the developed system’s performance ratio after stabilization. The paper demonstrated the effectiveness of the proposed approach in removing hair from dermoscopic images, which is a crucial preprocessing step in melanoma detection. However, the approach is dependent on the quality of the input images, and low-quality images may negatively impact the performance.

Choudhary et al. [72] proposed a neural-network-based method to separate dermoscopic images including two different kinds of skin lesions. The initiative’s proposed solution was divided into four steps that included initial image processing, skin lesion segmentation, feature extraction, and DNN-based classification. With a median filter, image processing was the initial stage in removing any extra noise. The specific locations of the skin lesions were then segmented using Otsu’s image-segmentation method. The third stage involved further extraction of the skin lesion characteristics, which were retrieved utilizing the RGB color model, 2D DWT, and GLCM. The classification of the various types of skin diseases using a backpropagation deep neural network and the Levenberg–Marquardt (LM) generalization approach to reduce the mean-squared error was the fourth stage. The ISIC 2017 dataset was used to train and test the suggested deep learning model. With DNN, they were able to outperform other state-of-the-art machine learning classifiers with an accuracy of 84.45%.

Lembhe et al. [73] proposed a synthetic skin-cancer-screening method using a solution or sequence from visual LR images. To improve the image-processing and machine learning methods, a deep learning strategy on super-resolution images was applied. Convolutional neural network models such as VGG 16, ResNet, and Inception V3 can be accurately recreated using image super-resolution (ISR) techniques. This model was created with the help of the Keras backend, and it was evaluated using a sequence or solution from visual LR photos. To improve the altering layers of the neural networks utilized for training, a deep learning strategy on the picture super-resolution was applied. The convolutional neural network model’s ISIC accuracy dataset, which is publicly available, was used to build the model.

A novel hybrid extreme learning machine (ELM) and teaching–learning-based optimization (TLBO) algorithm was developed by Priyadharshini et al. [74] as a flexible method for melanoma detection. While TLBO is an optimization technique used to fine-tune the network’s parameters for enhanced performance, the ELM is a single-hidden-layer feed-forward neural network that can be trained rapidly and accurately. In contrast to earlier studies, the authors used the two methodologies to identify skin lesions as benign or malignant images, potentially increasing the accuracy of melanoma identification. However, the performance of the proposed method was only tested on a single dataset for skin cancer detection, which is a drawback of the paper. Evaluating the performance of the algorithm on additional skin cancer datasets should have been assessed by the authors to establish its practicality and robustness.

For the purpose of detecting melanoma skin cancer, Dandu et al. [75] introduced a unique method that combines transfer learning with hybrid classification. To increase the accuracy of melanoma detection, the authors developed a hybrid framework that uses pretrained deep learning models for segmentation and incorporates a hybrid classification technique. The development of a hybrid strategy that successfully combines transfer learning and classification approaches was one of the paper’s contributions. The authors increased melanoma detection accuracy by modifying a pretrained convolutional neural network for skin lesion segmentation and mixing hand-crafted features with segmented lesion features in the classification process. The proposed approach was evaluated in terms of accuracy, precision, and recall on a benchmark dataset. However, the paper did have certain limitations, where clinical validation is needed to evaluate the generalizability and dependability of the suggested strategy across a range of demographics and skin types. The paper might also used more-thorough arguments and justifications for the features used for the hybrid classification technique. Furthermore, the reproducibility and comprehension might be improved by a more-detailed explanation of the specific features used and their significance to melanoma diagnosis. Last, but not least, despite the paper’s promise of increased performance in comparison to current procedures, there was a lack of a thorough comparative analysis using cutting-edge techniques. Such an analysis would offer a more-thorough evaluation of the advantages and disadvantages of the suggested strategy in comparison to other pertinent methods.

In this section skin lesion detection using machine learning and deep learning were examined, and in Table 8 presented summary of all the prior studies discussed in this study and their performance also presented in Table 9.

## 6. Discussion

In this section, we delve into a detailed analysis of the key aspects explored in our survey paper related to skin lesion classification and detection. We focused on papers exclusively dedicated to classification tasks and those solely addressing detection challenges. Additionally, we investigated the relationship between skin lesion dataset modalities and the number of papers utilizing them. Furthermore, we examined how the distribution of papers varied concerning their publication years. Lastly, we explored the relationship between the types of datasets used and the number of papers employing them. By examining these critical factors, we aimed to gain a comprehensive understanding of the trends and developments in skin lesion research, shedding light on the prevailing research priorities and areas for potential future exploration.

The findings from our survey, as illustrated in Table 10, Table 11 and Table 12, revealed the primary research emphases observed in the papers under consideration. A significant portion of the papers focused on classification tasks, indicating the prevalence of studies aimed at categorizing and labeling various entities within the dataset. However, we also noted that a smaller subset of papers placed their emphasis on detection tasks, highlighting the interest in identifying specific objects or occurrences of interest within the data. Moreover, a notable number of papers took a more-comprehensive approach, addressing both classification and detection aspects in their research, reflecting the need for a holistic understanding and analysis of the data. Furthermore, a few papers delved even deeper, incorporating segmentation alongside classification and detection in their investigations. This integration allowed for the precise delineation and localization of specific regions or structures within the dataset, providing more-detailed insights and facilitating advanced analyses.

The variation in research foci across the surveyed papers emphasized the multidimensional nature of the field, where researchers employed various methodologies and techniques to address distinct aspects of a dataset. The diversity of approaches contributes to a richer understanding of the datasets’ complexities and enables the development of robust algorithms and models to tackle real-world challenges effectively. As the field continues to advance, these findings offer valuable guidance for researchers seeking to identify potential research gaps and align their studies with the evolving trends and needs of the domain.

In this comprehensive survey paper, we performed a thorough collection of research papers published between the years 2017 and 2019. The content extracted from these papers primarily focused on their Section 1, making up approximately 11.11% of the total papers included in our analysis. By delving into these introductory sections, we aimed to gain insights into the prevalent themes, background knowledge, and contextual information used by researchers in their respective studies.

Notably, the majority of the papers we examined were relatively recent, with a substantial portion published in the year 2020, constituting around 6.18% of the papers in our survey. This suggests a growing interest and significant advancements in the research field during that particular year. The influx of publications in 2020 indicates an active and dynamic research landscape, with scholars contributing new perspectives and findings to the body of knowledge.

Moreover, we observed a substantial increase in publications in the subsequent years, with 2021 contributing to 11.11% of the papers. This steady growth indicates a sustained momentum in research activities, as researchers continued to investigate and explore various topics and areas of interest.

The year 2022 saw a remarkable surge in scholarly output, covering an impressive 38.27% of the papers in our survey. This surge may reflect emerging trends, breakthroughs, or significant developments in the field, garnering substantial attention from researchers and leading to a spike in academic contributions.

Even though the year 2023 was still ongoing at the time of our survey, it already showcased a notable presence, accounting for 33.33% of the papers. This suggests that research endeavors were thriving, and the year holds promise for numerous new discoveries and advancements.

By carefully analyzing the distribution of publications across these years, our survey paper provides a snapshot of the research landscape’s temporal evolution (Figure 2). The higher concentration of recent papers highlights the dynamic nature of the field and the continuous drive to explore new avenues and challenges. Moreover, it points to the significance of staying up-to-date with the latest research findings and integrating the most-current knowledge into ongoing studies.

Furthermore, our survey contributes to understanding the trends and areas of focus within the research community over time. The increasing trend in publications from 2020 to 2023 indicates that the topics being studied were of great interest to researchers, likely due to their relevance and potential impact on the broader scientific and practical domains.

In our systematic review paper, we conducted an in-depth analysis of a diverse range of skin lesion datasets, specifically focusing on the imaging modalities employed to capture the characteristics of these lesions. Our investigation yielded valuable insights into the distribution and prevalence of different imaging modalities within these datasets, as presented in Figure 3.

A significant portion, accounting for 48.12% of the datasets, utilized the dermoscopic image modality. Dermoscopy, a non-invasive imaging technique, plays a crucial role in dermatology and skin lesion research. It involves the use of a specialized dermatoscope, which is a handheld magnifying device with a light source, to examine the skin lesions at a higher level of magnification. Dermoscopic images provide clinicians and researchers with enhanced visualization of the morphological structures and patterns within the skin lesions, aiding in more-accurate diagnosis, classification, and monitoring of various skin conditions. The prominence of dermoscopic imaging in nearly half of the datasets underscores its importance as a preferred and highly informative modality in the field.

Another significant imaging modality, observed in approximately 33.33% of the datasets, was the macroscopic imaging modality. Macroscopic images are captured using conventional visible light photography, which allows for a comprehensive view of the skin lesions as perceived by the naked eye. While macroscopic images lack the fine details provided by dermoscopy, they offer a practical and easily accessible means of documenting skin lesions. These images are particularly useful in a clinical setting where dermoscopes might not be readily available, and they provide essential information about the external appearance and overall presentation of the skin lesions. Moreover, macroscopic images often serve as valuable complements to dermoscopic images, providing a broader context for the lesion’s evaluation.

In addition to dermoscopic and macroscopic imaging modalities, the remaining 18.52% of the datasets encompassed various other types of imaging modalities. These may include confocal microscopy, ultrasound imaging, multispectral imaging, or combinations of multiple imaging techniques. Each of these alternative modalities offers unique benefits and insights into specific aspects of skin lesions, enabling a comprehensive understanding of their underlying structures and pathological features. The inclusion of these diverse imaging modalities in a portion of the datasets indicates the continuous exploration and experimentation within the scientific community to advance the capabilities of skin lesion analysis and diagnosis.

In our systematic review paper, we conducted an extensive analysis of various research studies in the field of skin lesion detection, classification, and segmentation. As illustrated in Figure 4, we observed the utilization of different datasets in these studies. Notably, the HAM10000 dataset was employed in 38.02% of the papers, indicating its widespread adoption among researchers. The PH2 dataset, on the other hand, was found to be used in 8.50% of the papers. Although its usage was less prevalent compared to HAM10000, it still played a significant role in contributing to the body of knowledge in this area. Furthermore, we observed that the ISIC dataset was utilized in 33.8% of the research papers. The high usage of the ISIC dataset can be attributed to its large and diverse collection of skin lesion images, making it a valuable resource for developing and evaluating skin-lesion-detection and -classification algorithms. In addition to the three major datasets mentioned above, we discovered that the remaining datasets collectively covered 19.72% of the studies. These datasets might be more-specialized or domain-specific, serving specific research purposes, or comparatively smaller in size. Overall, the data from our systematic review indicated that the HAM10000, ISIC, and PH2 datasets were the most-commonly used and influential resources in the domain of skin lesion research. Researchers have heavily relied on these datasets to train, test, and benchmark their algorithms due to their richness, diversity, and representativeness of real-world skin lesions. By understanding the prevalence and usage of these datasets, we gain valuable insights into the current trends and directions in skin lesion research, allowing for better benchmarking and comparison of novel approaches in the field. It also highlights the need for continued efforts in curating and sharing high-quality datasets to further advance the state-of-the-art in skin lesion detection, classification, and segmentation.

## 7. Open Challenges for Skin Lesion Classification and Detection

Skin disease diagnosis is an area of ongoing research and development, with several open challenges that researchers and clinicians are actively addressing. Here are some of the key challenges in skin disease diagnosis, along with possible citations for further reading:Image analysis: It is still difficult to develop reliable automated image analysis methods for the detection of skin diseases. This entails the recognition, categorization, and segmentation of skin lesions from dermatoscopic or image-based data [76,77].Data standardization and annotation: The lack of standardized and annotated datasets for skin diseases hinders the development and evaluation of algorithms. Creating comprehensive datasets with accurate annotations is crucial for training and validating machine learning models [78,79,80].Interpretable decision support systems: Skin disease diagnosis often requires interpretability to gain trust from clinicians. Developing decision support systems that provide transparent explanations for their predictions is a challenge that needs to be addressed [81].Incorporating clinical data: Integrating patient history, symptoms, and other clinical data along with visual information can improve the accuracy of skin disease diagnosis. However, effectively utilizing heterogeneous clinical data remains an open challenge [76].Real-time diagnosis: Enabling real-time skin disease diagnosis in clinical settings is another challenge. Developing fast and efficient algorithms that can provide quick and accurate assessments is crucial for improving patient outcomes [82].Addressing bias in dermatological datasets: Many dermatological datasets suffer from biases, including under-representation of certain skin types and diseases. Overcoming these biases is essential to ensure fairness and accuracy in skin-disease-diagnosis algorithms [83].Augmenting small and imbalanced datasets: Obtaining large and balanced datasets for training skin-disease-diagnosis models can be challenging. Developing effective data-augmentation techniques and strategies to handle imbalanced classes is crucial for improving model performance [84].Explainability and interpretability: Interpreting the decisions made by skin-disease-diagnosis models is important for gaining trust and acceptance from healthcare professionals. Developing explainable and interpretable models that can provide insights into the decision-making process is an ongoing challenge [85,86].Generalization to external data: Ensuring the generalizability of skin-disease-diagnosis models to external datasets and real-world clinical settings is crucial. Models need to be robust enough to handle variations in imaging conditions, patient demographics, and disease presentations [87].Integration with clinical workflows: Seamlessly integrating skin disease diagnosis algorithms into clinical workflows poses a challenge. The development of user-friendly interfaces and systems that can assist healthcare professionals in real-time diagnosis is essential for practical implementation [88,89].Ethical issues associated with AI: Currently, all doctors and users of AI products face the ethical challenges brought on by this technology. As most of us know, artificial intelligence may greatly aid in diagnosing and classifying diseases such as dermatological and other conditions. However, it also contributed to the current methods of skin-related disease detection and treatment, which indeed raise severe ethical and dermatological questions. As a result, the AI research community has been inspired to concentrate on trustworthy and responsible AI research [77].Skin condition similarities: One of the most-common challenges in skin disease/cancer classification and detection is that many skin conditions have similarities between them that are not distinguishable visually [11].

These challenges highlight the ongoing research and development efforts in skin disease diagnosis, focusing on data biases, interpretability, generalization, practical integration into clinical settings, etc. Researchers continue to work towards addressing these challenges to improve the accuracy and usability of diagnostic tools for dermatological conditions.

## 8. Conclusions

This survey on classification, segmentation, and detection of skin diseases and skin cancer has brought to light the impressive developments in dermatology made possible by the application of artificial intelligence (AI) techniques. This survey paper demonstrated the potential of AI-based systems to increase diagnostic precision, boost patient outcomes, and completely transform the identification and management of skin disorders including cancer.

Traditional machine learning algorithms, deep learning algorithms, and image analysis methods have all been used by researchers to create complex models that can analyze dermatological images captured using different imaging modalities with high levels of accuracy, sensitivity, specificity, and F1-scores. These simulations have demonstrated their capacity to categorize different skin conditions and locate malignant tumors, matching and occasionally even outperforming the performance of professional dermatologists.

The application of AI to dermatology has enhanced patient care by creating new opportunities for more-precise diagnosis. In situations where access to dermatologists may be limited, the use of computer-aided diagnostic (CAD) systems has the potential to help healthcare practitioners make decisions. These solutions can help with triage, offer second views, and increase the effectiveness of clinical workflows, all of which will ultimately enhance patient care and results.

However, despite the enormous progress made, difficulties still exist in the creation and application of AI-based systems for the diagnosis of skin conditions and skin cancer. To ensure trustworthy and moral applications in clinical settings, concerns including the quality and diversity of training datasets, class imbalance, and the interpretability of AI models must be addressed. Additionally, careful consideration of data protection, regulatory compliance, and physician acceptability is necessary for the integration of these technologies into the current healthcare infrastructure. Future studies should concentrate on overcoming these difficulties and enhancing the precision and durability of AI-based skin disease classification, segmentation, and detection systems. The creation of explainable AI models should also be prioritized since they can promote transparent decision-making and foster a relationship of trust between healthcare professionals and AI systems.

In conclusion, the systematic review report has shown how the field of dermatology could be profoundly affected by AI technologies. We can anticipate additional developments in skin illness and skin cancer analysis with continuing study, development, and collaboration between AI experts and dermatologists. These developments promise to increase diagnostic precision, create tailored treatment regimens, and improve patient care, all of which will improve the management of dermatological disorders.

## Figures and Tables

**Figure 1 diagnostics-13-03147-f001:**
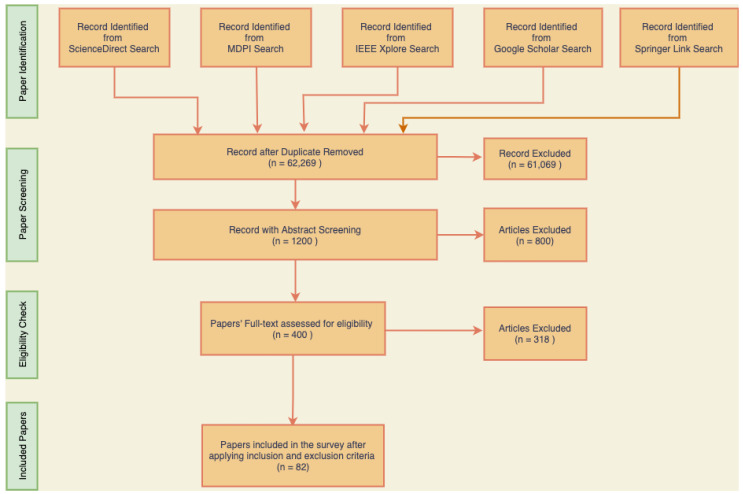
Flow diagram for study papers’ search and selection strategy from different databases.

**Figure 2 diagnostics-13-03147-f002:**
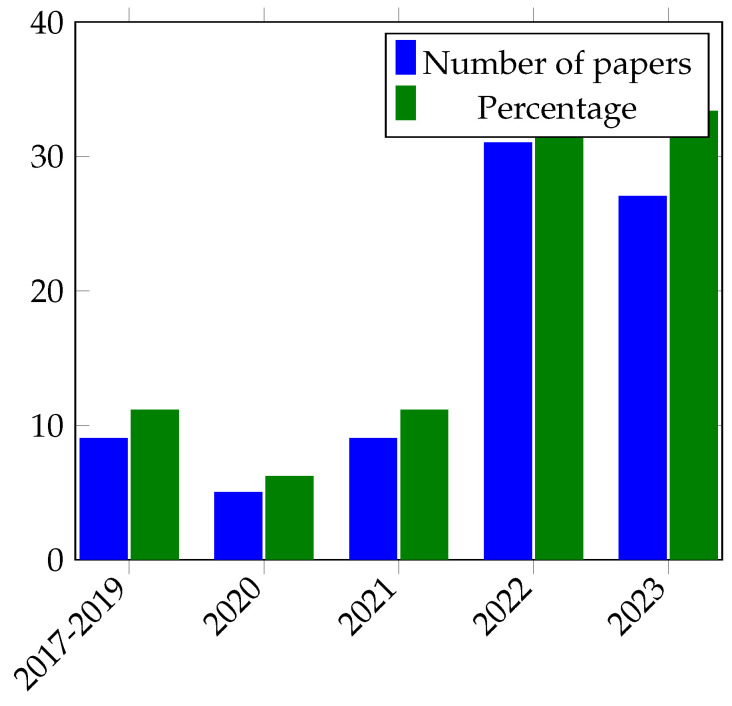
Reviewed papers’ distribution by year of publication.

**Figure 3 diagnostics-13-03147-f003:**
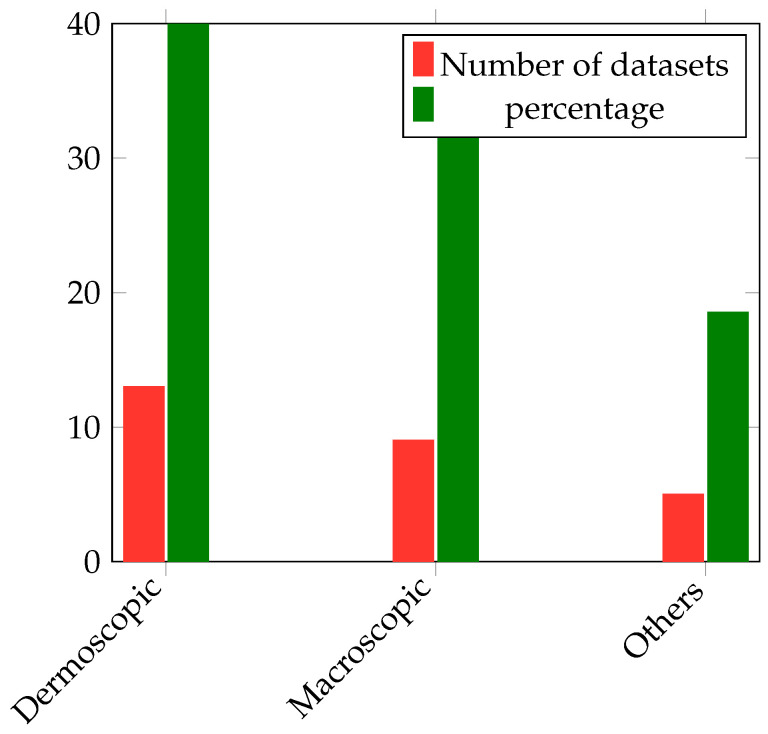
Imaging modalities in skin datasets.

**Figure 4 diagnostics-13-03147-f004:**
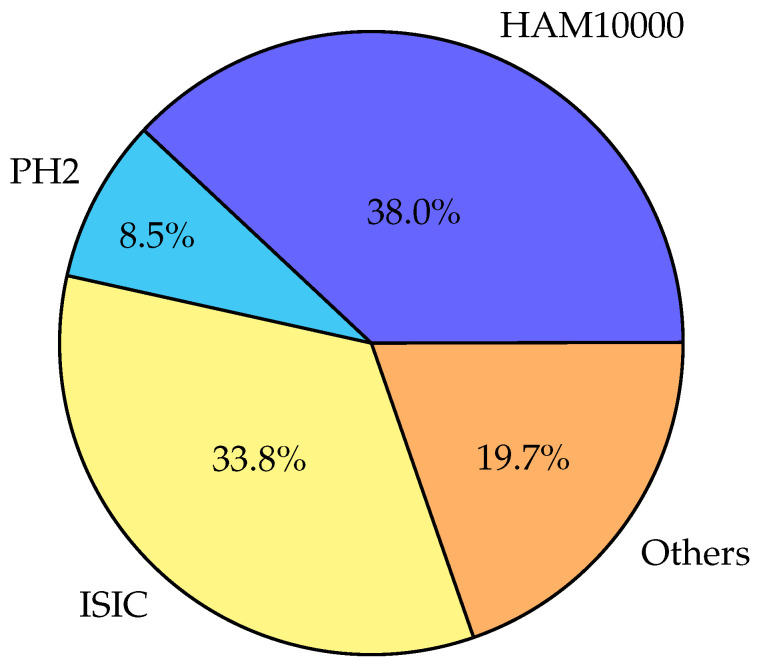
Datasets’ distribution in the systematically reviewed papers.

**Table 1 diagnostics-13-03147-t001:** Summary of related works with their contributions and limitations.

Author and Year of Publication	Contribution	Limitation
Grignaffini et al. [16], 2022	It reported a systematic literature review of recent research on the use of machine learning to classify skin lesions. The paper discussed datasets that are used commonly in skin lesion detection and classification.	Explaining the basic concepts of traditional ML algorithms is not important for the readers of such papers. Explaining the basic concepts of DL is also not important to readers that this paper targets.
Zafar et al. [19], 2023	A thorough assessment of the literature on the methods, procedures, and approaches used to examine skin lesions was made in this paper.	Research publications that analyzed skin lesions based on their complicated and uncommon features were excluded.
Hauser et al. [20], 2022	A systematic review of explainable artificial intelligence (XAI) in skin cancer recognition was made in this paper.	Limited to only XAI, it may have not covered all aspects of AI in skin cancer recognition. It did not compare XAI with others traditional ML and DL techniques in terms of various evaluation parameters.
Jeong et al. [21], 2023	Provided a thorough overview of dermatology in the literature review. A summarized review report of the current state of the datasets, transfer learning strategies, difficulties, and restrictions within the body of existing AI work was presented.	The survey was limited to papers published between 2015 and 2021. It overlookedpertinent studies.
Hasan et al. [22], 2023	The authors revealed that the ISIC is the most-commonly applied dataset in skin disease segmentation and classification. The survey enables researchers to determine the best experimental setup for skin lesion diagnosis.	Most of the publications (100+) considered in the papers were those published 6 years, between 2011 and 2016.
Bhatt et al. [23], 2022	Detailed analysis of cutting-edge machine learning methods for the identification and categorization of melanoma skin cancer.	Primarily focused on conventional machine learning methods for the identification and classification of melanoma skin cancer.
Mohammed and Al-Tuwaijari [24], 2022	Provided a summary of various classification systems for skin diseases based on machine learning methods.	Did not offer a methodical assessment of the approaches and did not include all of them.
Mazhar et al. [25], 2023	Reviewed the pertinent literature on skin cancer detection and the application of artificial intelligence (AI) in healthcare.	Ethical issues surrounding the use of AI in healthcare, particularly in relation to patient data security and privacy, may have been added to the article.

**Table 2 diagnostics-13-03147-t002:** Researchquestions for the systematic review work.

No.	Research Question	Objectives
1	What are the major targets of applying machine learning techniques in skin disease diagnosis?	To investigate the major targets of applying traditional machine learning and deep learning approaches for skin disease diagnosis.
2	What machine learning techniques are used in skin disease diagnosis?	To identify the commonly and recently proposed traditional machine learning techniques and deep learning techniques for skin disease diagnosis.
3	What are the common available dataset for skin disease diagnosis?	To identify the publicly available dataset that researchers can download either by online registration or freely available with or without payment.
4	How successful are the proposed machine learning techniques in skin disease diagnosis?	To analyze and compare the proposed machine learning techniques in the diagnosis of skin disease.
5	What are the future challenges in applying machine learning techniques for skin disease diagnosis?	To investigate the open questions in the application of machine learning techniques for skin disease diagnosis.

**Table 3 diagnostics-13-03147-t003:** Inclusion and exclusion criteria for paper selection.

Inclusion Criteria (ICs)	Exclusion Criteria (ECs)
IC1: The papers should focus on skin disease or cancer detection or segmentation or classification	EC1: Publications that are not focused on skin disease and cancer detection or classification or segmentation.
IC2: The papers should include different types of diseases, mainly skin cancer or melanoma.	EC2: Publications not peer-reviewed, abstracts, editorial letters, book reviews, and scientific reports.
IC3: The papers should be published in reputable journals with an impact factor and indexed in the Web of Science or Scopus or recognized conference proceedings.	EC3: M.Sc. and Ph.D. theses, posters, and seminars.
IC4: The studies should be written in English.	EC4: Studies that are published prior to 2020 except for Section 1 and Section 4.
IC5: Publication year for Section 1 and Section 4 can be any year of publication.	EC5: Skin disease detection and classification based on case studies.
IC6: The publication year for study paper to be included in the systematic review must be between 2020 and 2023	EC6: Study papers that are not peer-reviewed and journals that are not indexed in the Web of Science or Scopus.

**Table 4 diagnostics-13-03147-t004:** Summary of openly accessible datasets—ISIC Archive [26,27,28].

Dataset	Collection Site	Publication Year	Imaging Modality	Number of Category	Number of Images
ISIC2020	Hospital Clinic Barcelona	2020	Dermoscopic	2	7311
ISIC2020	University of Queensland	2020	Dermoscopic	-	8449
ISIC2020	Medical University Vienna	2020	Dermoscopic	2	4374
ISIC2020	Memorial Sloan Kettering Cancer Center	2020	Dermoscopic	5	11,108
ISIC2020	Sydney Melanoma Diagnosis Centre and Melanoma Institute Australia	2020	Dermoscopic	8	1884
HAM10000	Medical University of Vienna and skin cancer practice of Cliff Rosendahl in Queensland	2018	Dermoscopic	8	10,015
BCN20000	Hospital Clinic Barcelona	2019	Dermoscopic	9	19,424
JID	*Journal of Investigative Dermatology*	2018	Macroscopic	3	100
MSK 1-5	Memorial Sloan Kettering Cancer Center	2015 and 2017	Dermoscopic	15	3918
UDA	Google Research, Brain Team, and Carnegie Mellon University	2014 and 2015	Dermoscopic	7	617

**Table 5 diagnostics-13-03147-t005:** Summary of openly accessible datasets—ISIC Challenge test set and non-ISIC datasets [26,27,28].

Dataset	Collection Site	Publication Year	Imaging Modality	Number of Category	Number of Images
ISIC2020	Test set	2020	Dermoscopic	-	10,982
ISIC2019	Test set	2018 and 2019	Dermoscopic	-	8238
ISIC2018	Test set	2018	Dermoscopic	-	1000
PAD-UFES-20	Non-ISIC set	2020	Macroscopic	6	2298
PH2	Dermatology Service of Pedro Hispano Hospital	2013	Dermoscopic	3	200
7-point criteria evaluation database	Dr. Giuseppe Argenziano	2018	Dermoscopic and Macroscopic	15	2013
MED-NODE	University Medical Center Groningen (UMCG)	2015	Macroscopic	2	170
SKINL2	Instituto de Telecomunicações Campus Universitário de Santiago	2019	Light field photographs and dermoscopic photographs	8	814
SNU	University of Waterloo	-	Macroscopic	2	206
SDN-260	-	2019	Macroscopic	260	20,600

**Table 6 diagnostics-13-03147-t006:** Summary of non-openly accessible datasets, but downloaded upon request [26,27,28].

Dataset	Collection Site	Publication Year	Imaging Modality	Number of Category	Number of Images
Asan	Asan Institutional	2017	Macroscopic	12	17,125
Hallym	Asan Institutional	2017	-	1 152	
DERMOFIT image library	University of Edinburgh	-	-	10	1300
IMA205	-	2018	-	-	-
MoleMapper app patient photo	Ph.D. cancer biologist, Dan Webster	2017	Macroscopic	2	2422
SNU	University of Edinburgh	2018	Macroscopic	134	2201
Severance	-	2020	Macroscopic	43	40,331
Papadakis	-	2021	Macroscopic	1	156

**Table 7 diagnostics-13-03147-t007:** Datasets and associated download websites [26,27,28] accessed on 20 July 2023.

Dataset	Website
Monkeypox-dataset-2022	https://github.com/mahsan2/Monkeypox-dataset-2022
ACNE04	https://github.com/xpwu95/LDL
BCN_20000 (part of ISIC 2019)	https://challenge2019.isic-archive.com/data.html
Asan and Hallym	https://figshare.com/articles/Asan_and_Hallym_Dataset_Thumbnails_/5406136
Atlas Dermatologico	http://www.atlasdermatologico.com.br
DermQuest	http://dermquest.com
DermAtlas	http://www.dermatlas.net
Dermis	http://www.dermis.net/dermisroot/en/home/index.htm
Dermnet	http://www.dermnet.com
DermWeb	http://www.dermweb.com
Dermnet NZ	https://www.dermnetnz.org
Dermatoweb	http://www.dermatoweb.net
Danderm	http://www.danderm-pdv.is.kkh.dk/atlas/index.html
Dermatologia Praktyczna	http://derma.pl/
DermSynth3D	https://github.com/sfu-mial/DermSynth3D
DermX (525 dermatological images with diagnoses and diagnosis explanations by three dermatologists)	https://github.com/ralucaj/dermx
Dermofit	https://licensing.edinburgh-innovations.ed.ac.uk/i/software/dermofit-image-library.html
Derm7pt	http://derm.cs.sfu.ca/
Diverse Dermatology Images (DDI)	https://ddi-dataset.github.io
ENriching Health data by ANnotations of Crowd and Experts (ENHANCE): ABC criteria annotations of ISIC 2017 and PH2 datasets	https://github.com/raumannsr/ENHANCE
Fitzpatrick17k (16,577 clinical images with diagnosis and Fitzpatrick scale labels)	https://github.com/mattgroh/fitzpatrick17k
HAM10000	https://www.nature.com/articles/sdata2018161
Hellenic Derm Atlas	http://www.hellenicdermatlas.com/en
ISIC	https://isic-archive.com/
Islam et al. Monkeypox Skin Image Dataset 2022	www.Kaggle.com/datasets/arafathussain/monkeypox-skin-image-dataset-2022
MedMNIST	https://medmnist.com
Med-Node	http://www.cs.rug.nl/~imaging/databases/melanoma_naevi/
Meddean	http://www.meddean.luc.edu/lumen/MedEd/medicine/dermatology/melton/atlas.htm
MoleMap	https://molemap.co.nz
MSK	https://arxiv.org/abs/1710.05006
PH2	https://www.fc.up.pt/addi/ph2%20database.html
PAD-UFES-20 (clinical skin lesion images from smartphones)	https://data.mendeley.com/datasets/zr7vgbcyr2/1
Skin3D	https://github.com/jeremykawahara/skin3d
SD198	https://drive.google.com/file/d/1YgnKz3hnzD3umEYHAgd29n2AwedV1Jmg/view
Skin Cancer Detection	https://uwaterloo.ca/vision-image-processing-lab/research-demos/skin-cancer-detection
SKINCON	https://skincon-dataset.github.io/index.html
University of Iowa Clinical Skin Disease Images	http://www.medicine.uiowa.edu/dermatology/diseaseimages/
UWaterloo Skin Cancer Detection dataset (images taken from DermIS and DermQuest along with lesion segmentation)	https://uwaterloo.ca/vision-image-processing-lab/research-demos/skin-cancer-detection
XiangyaDerm	http://airl.csu.edu.cn/xiangyaderm/

**Table 8 diagnostics-13-03147-t008:** Summary of previous studies on skin disease classification with their performance.

Author/Year	Method	Dataset	Dataset Size	Acc (%)	Sn (%)	Sp (%)	R (%)	P (%)	F1 (%)
Ali et al. [30]	EfficientNets	HAM10000	10,015	87.9	88	88	88	88	87
Reddy et al. [36]	GWO	PH2	200	94.2	91.83	96.47	91.83	96.15	93.94
Inthiyaz et al. [66]	CNN	Xiangya-Derm	-	87	-	-	-	-	-
Srinivasu et al. [31]	DLNN + MobileNet V2 + LSTM	HAM10000	10,015	90.21	92.24	95.1	92.24	-	-
Shetty et al. [32]	ML + CNN	HAM10000	10,015	95.18	94	-	85	88	86
Wei et al. [34]	DenseNet + ConvNeXt	Peking-Union Medical College Hospital	2600	96.54	94.75	-	94.74	95.45	95.03
Wei et al. [34]	DenseNet + ConvNeXt	HAM10000	10,015	95.29	92.58	-	92.58	88.35	89.99
Almuayqil et al. [35]	DenseNet 201 + ML	HAM10000	10,015	99.94	91.48	98.82	91.48	97.01	-
Malibari et al. [37]	ODNNsingle bondCADSCC	-	-	99.90	-	-	-	-	-
Qian et al. [38]	DCNN-GMAB	HAM10000	10,015	91.6	73.5	96.4	73.5	-	-
Jain et al. [33]	OP-DNN	ISIC	23,906	95.6	91.2	97	91.2	92	-
Kumar et al. [15]	AUDNN	Kaggle + CIA	-	93.26	-	-	-	-	-
Nakai et al. [39]	EDBTM	HAM10000	10,015	95.84	-	-	-	96.1	-
Nakai et al. [39]	EDBTM	ISIC2017	-	92.1	90.1	91.9	-	-	-
Hossain et al. [40]	Customized ResNet50	EM + HAM10000	-	84.42 ± 1.36	87.93 ± 1.47	80.65 ± 3.9	-	83.1 ± 2.49	-
Hossain et al. [40]	Lightweight EfficientNetB0	EM + HAM10000	-	83.13 ± 1.2	85.21 ± 3.91	80.89 ± 2.95	-	82.83 ± 1.75	-
Afza et al. [41]	Hierarchical: NB	ISBI2016	1279	91.1	91	-	-	91.5	-
Afza et al. [41]	Hierarchical: NB	HAM10000	10,015	85.80	86	-	-	86.28	86.14
Afza et al. [41]	Hierarchical: NB	PH2	200	95.40	95.1	-	-	95.33	95.21
Alam et al. [42]	S^2^C-DeLeNet	HAM10000	10,015	91.03	90.58	90.58	90.58	90.38	90.48
Elashiri et al. [43]	HSBSO-LSTM	PH2	200	93.5	93.8	93.3	-	90.4	9.2
Elashiri et al. [43]	HSBSO-LSTM	HAM10000	10,015	93.8	93.9	93.8	-	33.9	49.8
Hsu and Tseng [45]	HAC-LF	ISIC2019	-	87.1	84.2	88.9	-	-	-
Omeroglu et al. [48]	Soft-attention-based multi-modal DL	7-point criteria evaluation (SPC)	1011	83.04	72.9	88.03	78.13	-	-
Serte and Demirel [49]	ResNet-18-based I-A1-H-V	ISIC2017	2000	81.5	-	97.5	-	-	-
Serte and Demirel [49]	ResNet-50-based I-A1-A2-A3	ISIC2017	2000	81	-	99.5	-	-	-
Bansal et al. [50]	BHHO-S algorithm + linear SVM	HAM10000	88	89	89	-	86	-	-
Gutiérrez et al. [51]	SSTF statistical fractal signatures + LDA classifier (4 classes)	ISIC2019	25,331	87	63	89	-	65	-
Gutiérrez et al. [51]	SSTF statistical fractal signatures + LDA classifier (7 classes)	ISIC2019	25,331	88	41	92	-	46	-
Thanka et al. [52]	VGG16 + XGBOOST	ISIC	1416	99.1	99.4	98.8	-	-	-
Thanka et al. [52]	VGG16 + LightBGM	ISIC	1416	97.2	97.8	96.6	-	-	-
Brinker et al. [53]	Ensembles: 3-CNNs	-	-	90.33	88.88	91.77	-	-	-
Alenezi et al. [54]	WT-DRNNet (ReLu)	ISIC2017	2750	96.91	-	97.68	-	96.43	95.79
Alenezi et al. [54]	WT-DRNNet (PReLu)	ISIC2017	2750	96.91	97.68	-	96.43	95.79	-
Alenezi et al. [54]	WT-DRNNet (Sigmoid)	ISIC2017	2750	96.91	-	97.68	-	96.43	95.79
Alenezi et al. [54]	WT-DRNNet (Hardlim)	ISIC2017	2750	96.91	-	97.68	-	96.43	95.79
Alenezi et al. [54]	WT-DRNNet (ReLu)	HAM10000	10,015	95.73	-	98.80	-	95.84	93.44
Alenezi et al. [54]	WT-DRNNet (PReLu)	HAM10000	10,015	95.36	-	98.62	-	95.59	93.37
Alenezi et al. [54]	WT-DRNNet (Sigmoid)	HAM10000	10,015	93.19	-	98.00	-	93.20	89.82
Alenezi et al. [54]	WT-DRNNet (Hardlim)	HAM10000	10,015	92.14	-	97.61	-	91.82	87.45
Alhudhaif et al. [55]	Soft-attention-based CNN	HAM10000 (unbalanced)	10,015	69.75	-	-	-	-	-
Alhudhaif et al. [55]	Soft-attention-based CNN	HAM10000 (balanced-SMOTE)	46,935	-	-	96	96.14	-	95.86
Alhudhaif et al. [55]	Soft-attention-based CNN	HAM10000 (balanced-ADASYN)	46,999	-	-	94.29	94.71	-	94
Alhudhaif et al. [55]	Soft-attention-based CNN	HAM10000 (balanced-RandomOverSampler)	46,935	-	-	88.57	90.14	-	89.29
Huang et al. [56]	DBN-MEFOA	HAM10000	10,015	97.99	92.99	97.00	-	96.99	91.99
Kalpana et al. [57]	ESVMKRF-HEAO	HAM10000	10,015	97.4	95.9	96	-	96.3	97.4
Shi et al. [58]	CMCNN-whole-slide image (WSI)	-	504	82.68	-	-	-	-	-
Shi et al. [58]	AGCN-whole-slide image (WSI)	-	504	95.24	-	-	-	-	-
Rafay and Hussain [59]	EfficientSkinDis: fine-tuned EfficientNet-B2	Atlas Dermatology and ISIC	4910	74	-	-	-	-	-
Rafay and Hussain [59]	EfficientSkinDis: fine-tuned EfficientNet-B2	Atlas Dermatology and ISIC	45,912	87.15	-	-	-	-	-
Kalaiyarivu and Nalini [61]	CNN	Customized hand images	-	87.5	-	-	-	-	-
Kousis et al. [62]	DenseNet169	HAM10000	10,015	92.25	93.59	-	-	-	93.27
Ahmad et al. [63]	CNN-layered BLSTM	Customized	6454	91.73	91.83	98.77	-	-	-
Ahmad et al. [63]	CNN-layered BLSTM	HAM10000	10,015	89.47	88.33	97.17	-	-	-
Aijaz et al. [64]	CNN	Dermnet (301) + BFL NTU (172)	473	84.2	84.33	86	-	-	-
Aijaz et al. [64]	LSTM	Dermnet (301) + BFL NTU (172)	473	72.3	72.33	75.16	-	-	-
Benyahia et al. [65]	DenseNet201 + Cubic SVM	ISIC2019	-	91.71	-	96.4	92.04	84.82	86.82
Benyahia et al. [65]	DenseNet201 + Fine KNN	ISIC2019	-	92.34	-	96.38	92.75	85.22	86.96
Benyahia et al. [65]	DenseNet201 + Cubic SVM	PH2	-	99	-	-	-	-	-
Benyahia et al. [65]	DenseNet201 + Quadratic SVM	PH2	-	99	-	-	-	-	-
Yanagisawa et al. [46]	DeepLabv3+- CNN	NSDD	16,313	90	90	90	-	-	-
Maqsood and Damaševičius [60]	MC-SVM	HAM10000	10,015	98.57	93.89	96.37	-	-	94.98
Maqsood and Damaševičius [60]	MC-SVM	ISIC2018	98.62	93.24	97.98	-	-	95.98	-
Maqsood and Damaševičius [60]	MC-SVM	ISIC2019	93.47	84.34	87.53	-	-	88.67	-
Maqsood and Damaševičius [60]	MC-SVM	PH2	98.98	98.03	98.70	-	-	98.87	-

**Table 9 diagnostics-13-03147-t009:** Summary of previous studies on skin disease detection with their performance.

Author and Year	Method	Dataset	Dataset Size	Acc (%)	Sn (%)	Sp (%)	R (%)	P (%)	F1 (%)
Dwived et al. [67]	Fast R-CNN			-	-	90	-	-	-
Alam and Jihan et al. [68]	DL+ Image Processing	-	-	85.14	-	-	-	-	-
Wan et al. [69]	Fusion Network	HAM10000	10,015	92.01	-	89.53	-	-	88.94
Kumar and Vanmathi et al. [70]	U-Net + RP-Net	-	-	92.5	92.1	91.7	-	-	-
Suicmez et al. [71]	Hybrid CNN-Gradient Boost Classifier	HAM10000	10,015	99.4	99.4	-	99.4	99.4	99.4
Suicmez et al. [71]	Hybrid CNN-Machine Learning	ISIC 2020	-	100	100	-	100	100	100
Lembhe et al. [73]	VGG16: ISR	ISIC	70.17	69	-	-	68	73	-
Lembhe et al. [73]	ResNet: ISR	ISIC	86.57	87	-	-	87	87	-
Lembhe et al. [73]	Inception V3: ISR	ISIC	91.26	92	-	-	89	92	-
Priyadharshini et al. [74]	ELM- TLBO	Kaggle and DermIS	300	-	-	-	92.45	89.72	91.64
Dandu et al. [75]	Ensemble Classifier	SIIM ISIC	-	86.38	-	-	86.50	86.16	-

**Table 10 diagnostics-13-03147-t010:** Summary of previous studies those focused on skin disease classification.

Author	Method	Objective
Ali et al. [30]	EfficientNets	Classification
Reddy et al. [36]	GWO	Classification
Inthiyaz et al. [66]	CNN	Classification
Srinivasu et al. [31]	DLNN + MobileNet V2 + LSTM	Classification
Shetty et al. [32]	ML + CNN	Classification
Wei et al. [34]	DenseNet + ConvNeXt	Classification
Wei et al. [34]	DenseNet + ConvNeXt	Classification
Almuayqil et al. [35]	DenseNet 201 + ML	Classification
Malibari et al. [37]	ODNNsingle bondCADSCC	Classification
Qian et al. [38]	DCNN-GMAB	Classification
Jain et al. [33]	OP-DNN	Classification
Kumar et al. [15]	AUDNN	Classification
Nakai et al. [39]	EDBTM (Dataset: HAM10000)	Classification
Nakai et al. [39]	EDBTM (Dataset: ISIC2017)	Classification
Hossain et al. [40]	Customized ResNet50	Classification
Hossain et al. [40]	Lightweight EfficientNetB0	Classification
Afza et al. [41]	Hierarchical: NB	Classification
Afza et al. [41]	Hierarchical: NB (Dataset: PH2)	Classification
Afza et al. [41]	Hierarchical: NB (Dataset: HAM10000)	Classification
Alam et al. [42]	S^2^C-DeLeNet	Classification
Elashiri et al. [43]	HSBSO-LSTM (Dataset: PH2)	Classification
Elashiri et al. [43]	HSBSO-LSTM (Dataset: HAM10000)	Classification
Benyahia et al. [65]	DenseNet201 + Cubic SVM	Classification
Benyahia et al. [65]	DenseNet201 + Quadratic SVM	Classification

**Table 11 diagnostics-13-03147-t011:** Summary of previous studies those focused on skin disease detection.

Author	Method	Objective
Dwived et al.[67]	Fast R-CNN	Detection
Alam and Jihan et al. [68]	DL+ Image Processing	Detection
Wan et al.[69]	Fusion Network	Detection
Kumar and Vanmathi et al. [70]	U-Net + RP-Net	Detection
Suicmez et al. [71]	Hybrid CNN-Gradient Boost Classifier	Detection
Suicmez et al. [71]	Hybrid CNN-Machine Learning	Detection
Lembhe et al. [73]	VGG16: ISR	Detection
Lembhe et al.[73]	ResNet: ISR	Detection
Lembhe et al. [73]	Inception V3: ISR	Detection
Priyadharshini et al. [74]	ELM- TLBO	Detection
Dandu et al. [75]	Ensemble Classifier	Detection

**Table 12 diagnostics-13-03147-t012:** Summary of previous studies those focused on skin disease for multiple objectives.

Author	Method	Objective
Reddy et al. [36]	GAWO	Segmentation and Classification
Malibari et al. [37]	ODNNsingle bondCADSCC	Classification and Detection
Afza et al. [41]	Hierarchical: NB	Segmentation, Classification, and Detection
Afza et al. [41]	Hierarchical: NB	Segmentation, Classification, and Detection
Afza et al. [41]	Hierarchical: NB	Segmentation, Classification, and Detection
Alam et al. [42]	S^2^C-DeLeNet	Segmentation, Classification, and Detection
Maqsood and Damaševičius [60]	MC-SVM	Classification and Detection
Maqsood and Damaševičius [60]	MC-SVM	Classification and Detection
Maqsood and Damaševičius [60]	MC-SVM	Classification and Detection
Maqsood and Damaševičius [60]	MC-SVM	Classification and Detection
Dandu et al. [75]	Ensemble Classifier	Segmentation, Classification, and Detection
Yanagisawa et al. [46]	DeepLabv3+- CNN	Segmentation and Classification

## Data Availability

Not applicable.

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
