# Peer review of "Skin Lesion Classification and Detection Using Machine Learning Techniques: A Systematic Review"

_diagnostics, 2023, doi:10.3390/diagnostics13193147_

Round 1
Reviewer 1 Report
The publication presents a very important problem of skin lesion examination. In skin cancer, the most important thing is rapid diagnosis and observation of lesions, in order to quickly detect the disease and increase the patient's chances of survival. Therefore, this topic is important, especially because the incidence of skin cancer is increasing every year. The text is clear. The authors have summarized much of the data in tables and figures, which further clarifies the data. I believe that the publication is suitable for publication in the Journal.
Author Response
Dear Reviewer, Thank you for you for your support and I am very grateful for your professional contribution and technical feedback. With regards, AuthorReviewer 2 Report
The author is to commend for the enormous work. I have only some suggestions: to describe in more detail the methods of the systematic review, and for example, when selecting studies with a diagnostic test accuracy design, were the studies evaluated for their quality according to a quality tool commonly used in systematic reviews sch as QUADAS-2? And if not, explain clearly why. Then, did the author all the work alone? Given that a control in searching and extraction of data is part of a systematic review, this should be acknowledged.
Author Response
Dear Reviewer, Thank you for you for your support and I am very grateful for your professional contribution and technical feedback. With regards, Author
Reviewer 3 Report
This paper surveys existing works on skin image analysis. Plenty of information related to this research area including available datasets, associated clinical goals and methods used are discussed. It is a useful work for the newcomer to this field. However, it would have been better if the presentation was a bit better. I suggest showing the challenges in a pictorial manner (publicly available free image datasets can be considered). Second, a critical remark on how the evolution of machine learning impacting skin image analysis techniques will be a nice contribution. I find that Acne, Psoriasis ..etc diseases are mentioned but no work is considered for review. If so then discard in the introduction section also.
Some minor mistakes are there.
Author Response

(The authors gave the same response as above.)
